# SkillMimicGen: Automated Demonstration Generation for Efficient Skill Learning and Deployment

**Caelan Garrett**\*, **Ajay Mandlekar**\*, **Bowen Wen, Dieter Fox**
NVIDIA, \* equal contribution

**Abstract:** Imitation learning from human demonstrations is an effective paradigm for robot manipulation, but acquiring large datasets is costly and resource-intensive, especially for long-horizon tasks. To address this issue, we propose SkillMimicGen (SkillGen), an automated system for generating demonstration datasets from a few human demos. SkillGen segments human demos into manipulation skills, adapts these skills to new contexts, and stitches them together through free-space transit and transfer motion. We also propose a Hybrid Skill Policy (HSP) framework for learning skill initiation, control, and termination components from SkillGen datasets, enabling skills to be sequenced using motion planning at test-time. We demonstrate that SkillGen greatly improves data generation and policy learning performance over a state-of-the-art data generation framework, resulting in the capability to produce data for large scene variations, including clutter, and agents that are on average 24% more successful. We demonstrate the efficacy of SkillGen by generating over 24K demonstrations across 18 task variants in simulation from just 60 human demonstrations, and training proficient, often near-perfect, HSP agents. Finally, we apply SkillGen to 3 real-world manipulation tasks and also demonstrate zero-shot sim-to-real transfer on a long-horizon assembly task. Videos, and more at `https://skillgen.github.io`.

**Keywords:** Imitation Learning, Manipulation, Planning

## 1 Introduction

Imitation learning from human demonstrations is an effective approach for training robots to perform different tasks [1, 2]. One popular technique is to have humans teleoperate robot arms to collect datasets for tasks of interest and then subsequently use the data to train robots to perform these tasks autonomously [3, 4]. Recent efforts have demonstrated that large, diverse datasets collected by teams of human demonstrators result in impressive and robust robot performance, and even allow the robots to generalize to different objects and tasks [2, 5–8]. However, collecting large datasets in this way is costly and resource-intensive, often requiring multiple human operators, robots, and months of human effort. Acquiring datasets for challenging long-horizon tasks that require sequencing several manipulation behaviors together is even more difficult and costly [9].

The need for large datasets has motivated the development of data generation systems [10–12] that seek to produce task demonstrations with minimal human involvement. For example, some systems combine teleoperation and planning within the same demonstration, partially automating the demonstrating process, which ultimately allows a human to teleoperate several robots in parallel [13]. Alternatively, some systems further reduce human involvement through demonstration adaptation. For example, MimicGen [11], uses a small number of human task demonstrations to automatically generate large datasets by splitting the source human data into object-centric sequences of end-effector targets, and then selectively transforming and sequencing such segments in new settings. However, this and other naive strategies for composing human segments together can produce lower-quality demonstrations with unintended collisions in the environment, and have heterogeneous motions that are difficult for policy learning algorithms to learn from, especially in real-world settings.

8th Conference on Robot Learning (CoRL 2024), Munich, Germany.

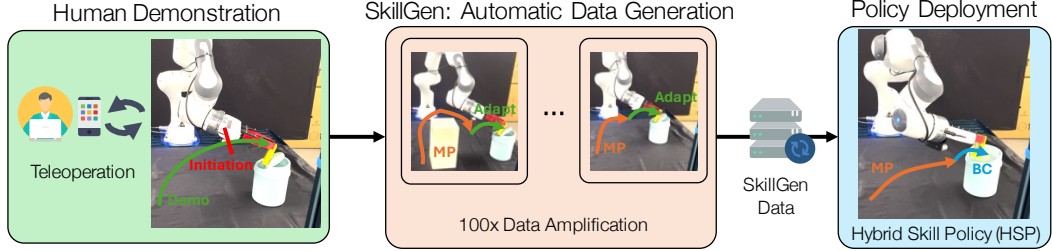

Figure 1: **SkillGen Overview.** SkillGen trains proficient agents with minimal human effort. (*left*) First, a human teleoperator first collects ∼ 3 demonstrations of the task and annotates the start and end of the skill segments, where each object interaction happens. (*middle*) Then, SkillGen automatically adapts these local skill demonstrations to new scenes and connects them through motion planning to amplify the number of successful demonstrations. (*right*) These demonstrations are used to train Hybrid Skill Policies (HSP), agents that alternate between closed-loop reactive skills and coarse transit motions carried out by motion planning.

We also seek to minimize the number of required human demonstrations but improve the flexibility and efficacy of adapted demonstrations. To that end, we first observe that control difficulty is often not uniformly spread across a task. Specifically, in order to solve many manipulation tasks, the robot must first move itself in free space in order to reach a state where it can manipulate the world through contact. For example, consider the cleanup task in Fig. 1. The robot must move through free space before picking the butter and also before inserting the butter into the trash can. This kind of free space motion can be easy for planning systems, and greatly reduce the burden on policy learning.

From this observation, we propose **SkillMimicGen (SkillGen)**, a system that leverages the notion of a manipulation *skill* to isolate demonstration adaptation to contact-rich segments. At data-generation time, SkillGen synthesizes candidate demonstrations by executing several adapted skill segments in sequence, connected through motion planning. At test-time, SkillGen not only learns control policies for these skills but also initiation and termination conditions, enabling them to be sequenced using planning in a similar manner but without any requirements regarding state observability.

**We make the following contributions:**
• We introduce SkillMimicGen (SkillGen), an automated system for generating demonstration datasets through decomposing tasks into motion segments and skill segments that are adapted from a few human demos.
• We propose a Hybrid Skill Policy (HSP) framework that learns skill initiation, control, and termination components, enabling skills to be combined in sequence at test time using motion planning.
• We show that SkillGen improves data generation and policy learning performance over an existing state-of-the-art data generation framework. Specifically, SkillGen is robust to large scene variation, such as clutter, and produces policies that on average are 24% more successful than MimicGen [11].
• We demonstrate the efficacy of SkillGen by generating 24K+ demonstrations from 60 human demonstrations across 18 task variants in simulation and training proficient, often near-perfect, high-performing HSP agents. Finally, we successfully apply SkillGen to 3 real-world manipulation tasks, and also demonstrate zero-shot sim-to-real transfer on a long-horizon assembly task.

## 2 Related Work

**Data Collection for Robotics.** Robot teleoperation [3, 4, 14–23] is a popular method for collecting task demonstrations – here, humans use a teleoperation device to control a robot and guide it through tasks. The robot sensor streams and control actions during operation are logged to a dataset. Several efforts [2, 5–8] have scaled this paradigm up by using a large number of human operators and robot arms over extended periods of time (*e.g.* months). Some works have also allowed for robot-free data collection with specialized hardware [24, 25], but human effort is still required for data collection. In contrast, SkillGen automatically generates data with just a handful of human demonstrations. Other works seek to generate datasets automatically using pre-programmed demonstrators in simulation [10, 26–31], but scaling these approaches to a larger variety of tasks can be difficult.

**Imitation Learning and Data Augmentation.** Behavioral Cloning (BC) [32] is a typical method for learning policies offline from demonstrations, and has been widely used in robot manipulation [3,

16, 27, 33–45]. Some works leverage offline data augmentation to increase the size of the training dataset for learning policies [1, 46–57]. Instead, SkillGen collects new datasets online.

**Imitation Learning with Hybrid Controllers.** SayCan [6] composes skills learned from demonstrations using a language model and learns when to begin and end each skill However, each skill starts when the previous one ends – in contrast, our learned skills are local manipulation behaviors and transit is carried out via motion planning. Other works [58–60] learn "keyframe" pose actions from demonstrations and execute them using motion planning, but they lack closed-loop control using learned policies. Some imitation learning methods decompose learning into coarse-grained and fine-grained motions [13, 61–64], but most use naive linear interpolation to carry out coarse-grained motions [61, 62], which is susceptible to collisions. Others [63–65] learn open-loop segments for fine-grained motions, instead of closed-loop skills like our methods. Wang *et al.* [66] learn parametric skills using Gaussian Processes and deploy them in a Task and Motion Planning (TAMP) [67] system. In HITL-TAMP [13], a TAMP planner decides when to employ an agent trained with imitation learning for skill segments; however, it is *TAMP-gated*, meaning that skill start and end conditions are engineered into the TAMP model instead of learned.

**MimicGen.** MimicGen [11] is a data generation system that takes a small source set of human demonstrations on a task and generates larger sets of demonstrations. It builds on replay-based imitation learning methods [65, 68–74], which address new task instances by adapting and replaying motion from existing human data. MimicGen segments the source demonstrations into a contiguous set of object-centric subtask segments. Then, given a new task instance, MimicGen transforms and replays open-loop subtask segments from the source data one-by-one to generate a new demonstration. However, because MimicGen naively stitches source demonstrations with linear interpolation, it can produce lower quality demonstrations that collide with the environment, and have heterogeneous motions difficult for policy learning. By instead adopting a skill-based framework, SkillGen avoids these pitfalls at data generation time and produces more robust behavior at deployment time.

## 3 Prerequisites

**Imitation Learning.** Each robot manipulation task is modeled as a Partially Observable Markov Decision Process (POMDP). We are given a dataset of $N$ demonstrations $\mathcal{D} = \{(s_0^i, o_0^i, a_0^i, s_1^i, o_1^i, a_1^i, ..., s_{H_i}^i)\}_{i=1}^N$ consisting of states $s \in \mathcal{S}$, observations $o \in \mathcal{O}$, and actions $a \in \mathcal{A}$. Each initial state $s_0^i \sim D$ is sampled from the initial state distribution $D \subseteq \mathcal{S}$. We aim to learn a robot control policy $\pi : \mathcal{O} \rightarrow \mathcal{A}$ that maps observation space $\mathcal{O}$ to a distribution over action space $\mathcal{A}$. Behavioral Cloning (BC) [32] is a common method to obtain such a policy – it uses optimization to find a policy that maximizes the likelihood of producing the data $\arg\max_\theta \mathbb{E}_{(s,o,a)\sim\mathcal{D}}[\log \pi_\theta(a \mid o)]$. In this work, we train policies via BC and combine them with various mechanisms to exchange control between a learned policy and a motion planner.

**Assumptions.** Similar to prior work [11], we make the following assumptions. **(A1):** The policy action space $\mathcal{A}$ consists of continuous pose commands for an end effector controller along with a discrete gripper command. This allows us to treat the actions in a human demonstration as a sequence of target poses for a task-space end-effector controller. **(A2):** The task involves a set of manipulable objects $\{O_1, ..., O_k\}$. **(A3):** During data collection, the pose of an object can be observed or estimated prior to the robot making contact with that object.

## 4 SkillMimicGen

We seek to learn visuomotor policies from demonstrations with minimal human effort by adapting a small number of human demonstrations to a large set of system states to facilitate automated demonstration generation. However, at both demonstration and deployment time, control difficulty is not uniformly spread across an episode. Specifically, in order to solve many manipulation tasks, the robot must first move itself in free space in order to reach a state where it can manipulate the world through contact. Free space motion can easily be carried out via motion planning and greatly reduce the policy learning burden. Thus, we propose decomposing tasks into *motion* and *skill* segments in order to isolate both demonstration generation and learning to just the skill segments, which will improve the quality of demonstrations and learned policies. We accomplish this by learning local manipulation *skills* that we combine in sequence using motion planning (Section 4.1). We show how

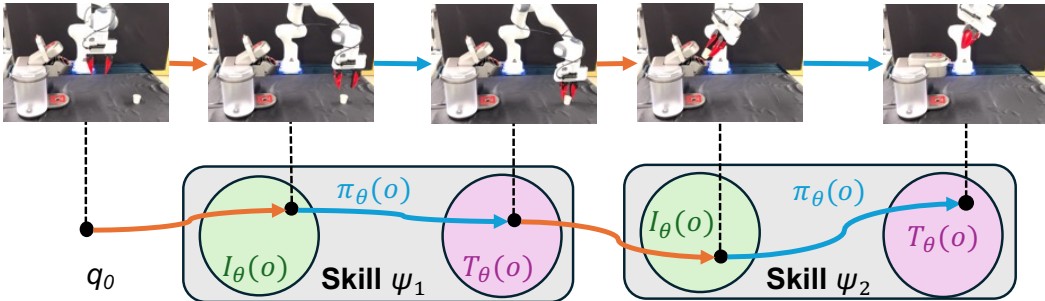

Figure 2: **HSP Deployment.** At test-time SkillGen, executes several learned skills in sequence, using motion planning to connect the termination state of the last skill with an initiation state of the next skill. Each skill consists of the initiation condition $I_\theta$, the closed-loop controller $\pi_\theta$, and the termination condition $T_\theta$.

adopting a skill-based framework allows for more focused demonstration replay (Section 4.4) and ultimately improved policy performance during deployment (Section 4.6).

## 4.1 Skills Framework

Building off of the *options* [75] formalism from reinforcement learning, we define a *skill* $\psi = \langle O, \mathcal{I}, \pi, \mathcal{T} \rangle$ as a tuple consisting of an *object* to be manipulated $O$, *initiation* condition $\mathcal{I}$, *policy* $\pi$, and a *termination* condition $\mathcal{T}$. The initiation condition $\mathcal{I}$ defines a set of states where control using policy $\pi$ can begin. The termination condition $\mathcal{T}$ defines a set of terminal states for policy $\pi$. We will use this skill abstraction to model all three phases of SkillGen, namely the initial teleoperation demonstrations (Section 4.3), the automated demonstration adaptation and amplification (Section 4.4), and the system execution at deployment time (Section 4.6).

## 4.2 Transit and Transfer Motion

Most tasks require performing multiple skills in sequence to complete them, such as the task in Fig. 2, which involves a *pick* skill to grasp the coffee pod and an *insert* skill load the pod in the coffee machine. In order to first reach the pick skill and then move the pod to the pod holder for the insert skill, the robot must perform two kinds of classical free-space motion [76, 77]. The first is *transit* motion, where the robot moves by itself without modifying the world. The second is *transfer* motion, where the robot is grasping an object approximately rigidly and transports the object as it moves. Thus, at both demonstration generation (Section 4.4) and system deployment (Section 4.6) time, SkillGen alternates between transit or transfer motion and manipulation skills.

SkillGen is a bilevel hierarchy where the skill initiation and termination induce the start and end robot configurations ($q$ and $q_*$) for the motion segments. Namely, the termination condition $\mathcal{T}_i$ from the prior skill $\psi_i$ governs the robot configuration $q$ prior to the motion, and the initiation condition $\mathcal{I}_{i+1}$ of the next skill $\psi_{i+1}$ defines the set of target end-effector poses $T_W^E \in \mathcal{I}_{i+1} \subseteq \mathrm{SE}(3)$, where $E$ is the end-effector frame and $W$ is the world frame. To generate these motions, we first convert task-space pose $T_W^E$ to joint-space configuration $q_*$ using inverse kinematics and then plan and execute a joint-space path from current configuration $q$ to $q_*$ with a motion planner.

## 4.3 Source Demonstrations

We assume a small *source* dataset of human demonstrations $\mathcal{D}_{\mathrm{src}}$ collected on the task and our aim is to automatically generate a large dataset $\mathcal{D}$ on either the same task or a task variant. We start by annotating each trajectory in the source dataset $\tau \in \mathcal{D}_{\mathrm{src}}$ with the start and end of each skill. This decomposes the demonstration into an alternating sequence of motion and skill trajectories $\tau = (\tau_{1m}, \tau_{1s}, ..., \tau_{Nm}, \tau_{Ns})$, where $\tau_{im}$ and $\tau_{is}$ denote motion and skill segments respectively. For source demonstrations provided by conventional teleoperation, these annotations can easily be annotated by a human. In our experiments, we choose to use demonstrations from the HITL-TAMP system [13], where the human only demonstrates local skill segments of each task, and the rest is handled by a TAMP system. In this case, annotations can be extracted automatically – each $\tau_{im}$ and $\tau_{is}$ is a TAMP and human segment respectively. Within each skill segment $\tau_{is}$, each end-effector pose action $T_W^{A_t}$ (Sec. 3, A1) is stored in the frame of skill object $O_i$ as $T_{O_i}^{A_t} \leftarrow (T_W^{O_i})^{-1} T_W^{A_t}$, where

$T_W^{O_i}$ is the pose of object $O_i$ observed prior to the skill. The first robot end effector pose in the skill demonstration $T_{O_i}^{E_0} \leftarrow \tau_{is}[0]$ is the *initiation state* and that will be the target end-effector pose for transit and transfer motion planning. The last pose in the demonstration $T_{O_i}^{E_K}$ implicitly defines the *termination state*, which will be learned through binary classification.

## 4.4 Demonstration Generation

The demonstrations $\mathcal{D}$ are generated through an automated trial-and-error process. Given a new initial state, SkillGen adapts existing skill segments to the new initial state and executes them in sequence with motion segments to generate a new demonstration. First, a reference skill segment $\tau_{is}$ is sampled. Next, the corresponding initiation state $T_{O_i}^{E_0}$ is used along with the pose $T_W^{O_i'}$ of object $O_i$ in the new scene to obtain an end-effector pose for where the new skill segment should start, $T_W^{E_0'} \leftarrow T_W^{O_i'} T_{O_i}^{E_0}$. Next, the reference skill segment, expressed as a sequence of end-effector pose actions, $\tau_{is} = (T_{O_i}^{A_0}, ..., T_{O_i}^{A_K})$ is transformed to $\tau_{is}' = (T_W^{A_0'}, ..., T_W^{A_K'})$ where $T_W^{A_t'} \leftarrow T_W^{O_i'} T_{O_i}^{A_t}$. This transformation preserves the new end-effector pose actions with respect to the object frame [11]. The new skill segment $\tau_{is}'$ is executed by the end-effector controller. The steps above repeat for each skill, and then SkillGen checks for task success and only keeps the demonstration if it was successful. Seed Appendix O for pseudocode displaying the demonstration generation process.

## 4.5 Initiation Augmentation

At test time, learned skills trained on the generated data will be responsible for predicting both initiation targets for the motion planner and skill segments by employing a closed-loop agent that decides when to terminate. However, small differences in target pose predictions as well as motion plan tracking errors can cause learned policies to start out-of-distribution, thus reducing their accuracy. To mitigate such issues, SkillGen optionally adds noise to initiation states $T_W^{E_0}$, producing new initiation states $T_W^{E_0'}$, during data generation to broaden the support of the initiation set. To account for changing the initiation state, we consequently plan a *recovery segment* at the start of $\tau_{is}'$, consisting of a sequence of pose actions that moves from new $T_W^{E_0'}$ pose to the original pose $T_W^{E_0}$. This ensures that the new initiation state $T_W^{E_0'}$ is connected to the demonstration segment $\tau_{is}'$ when training closed-loop skill policies. See Appendix G for full details.

## 4.6 Policy Learning

**Hybrid Skill Policy (HSP):** We learn *parameterized skills* $\psi_\theta = \langle O, \mathcal{I}_\theta, \pi_\theta, \mathcal{T}_\theta \rangle$ using the generated datasets (parameterized by $\theta$). The initiation condition $\mathcal{I}_\theta : \mathcal{O} \to \mathrm{SE}(3)$ is trained to predict initiation states $T_W^{E_0}$ from the last observation $o$ on the prior skill. The policy $\pi_\theta : \mathcal{O} \to \mathcal{A}$ is trained on direct observation and action pairs $\langle o, a \rangle$ with BC (see Sec. 5). The termination condition $\mathcal{T}_\theta : \mathcal{O} \to \{0, 1\}$ is a classifier that predicts whether the skill is at a termination state based on the most recent observation $o$. During task deployment (Fig. 2), for each skill $\psi_\theta \in \Psi$ in a given sequence of skills $\Psi$, SkillGen predicts the initiation state $T_W^{E_0'} \leftarrow \mathcal{I}_\theta(o)$, plans and executes a path to it using a motion planner, and rolls out the learned policy by predicting actions $a \leftarrow \pi_\theta(o)$ until $\mathcal{T}_\theta(o)$ predicts policy termination. Then, this process repeats with the next skill (pseudocode in Appendix O).

**HSP Variants:** We consider two approaches for learning initiation conditions $\mathcal{I}_\theta$: *HSP-Reg* and *HSP-Class*. HSP-Reg formulates learning as a regression problem and directly predicts an initiation pose from the last observation. HSP-Class frames learning as classification problem over the initiation states in the source dataset $\mathcal{D}_{\mathrm{src}}$, where the classifier predicts which source demonstration spawned the generated demonstration. Once classified, HSP-Class adapts the predicted initiation state to the current state using the pose adaptation procedure previously described in Section 4.4. However, recall that this requires the current pose $T_W^{O'}$ of object $O$, and thus HSP-Class assumes that object poses are known or can be estimated at the start of each skill segment. Ultimately, HSP-Class requires an additional observability assumption over HSP-Reg; however, this enables HSP-Class to perform discrete prediction over known pose candidates instead of continuous prediction over $\mathrm{SE}(3)$. Finally, we also consider *HSP-TAMP*, which deploys just the learned policies $\pi_\theta$ within HITL-TAMP [13], without the learned initiation and termination conditions.

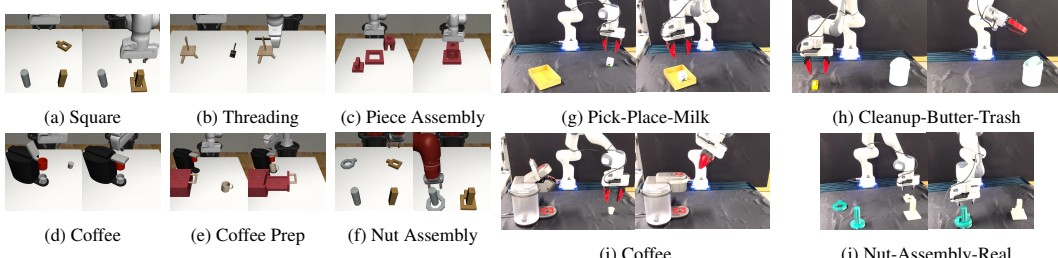

| (a) Square | (b) Threading | (c) Piece Assembly | (g) Pick-Place-Milk | (h) Cleanup-Butter-Trash |
| (d) Coffee | (e) Coffee Prep | (f) Nut Assembly | (i) Coffee | (j) Nut-Assembly-Real |

Figure 3: **Tasks.** We deploy SkillGen on 6 simulation tasks (18 task variants, see Appendix J) (a-f) and 4 real-world tasks (g-j). These tasks involve fine-grained insertion (a-d), composing several manipulation behaviors together (e, f), real-world data generation and training (g-i) and zero-shot sim-to-real policy transfer (j).

## 5 Experiment Setup

**Tasks and Task Variants.** We applied SkillGen to a broad range of tasks (see Fig. 3, full details in Appendix J) and task variants. Each task has a nominal reset distribution ($D_0$), and broader, more challenging reset distributions ($D_1$, $D_2$) [11]. All simulation tasks are implemented in robosuite [78] using its MuJoCo backend [79]. We experiment on simulated **Fine-Grained Tasks** (Square, Threading, Coffee, Piece Assembly) that require insertion, pulling, and pushing as well as **Long-Horizon Tasks** (Nut Assembly, Coffee Prep) that require chaining multiple behaviors together. Additionally, we experiment on **Real-Robot Tasks** (Pick-Place-Milk, Cleanup-Butter-Trash, Coffee), and **Sim-to-Real Tasks** (Nut-Assembly-Sim, Nut-Assembly-Real) to investigate SkillGen's propensity for zero-shot sim-to-real policy deployment.

**Data Generation and Imitation Learning.** For most of the experiments, a source dataset of 10 demonstrations was collected for each task on the $D_0$ variant by a single human operator using the HITL-TAMP teleoperation system [13]. SkillGen was used to generate 1000 successful demonstrations for each task variant ($D_0$, $D_1$, $D_2$) (see Appendix J for details), using each task's source dataset. Motion augmentation (Sec. 4) is only used to generate data to train HSP-Reg agents; HSP-TAMP and HSP-Class agents are trained on datasets generated without motion augmentation. See Appendix H for full policy learning details. The agent control policies used in the hybrid control policies ($\pi_\theta$) were trained using BC with an RNN architecture [1] with the same hyperparameters from MimicGen. Policy performance is reported as the maximum success rate across all policy evaluations as in Mandlekar *et al.* [1]. All agents are trained with front-view and wrist-view RGB observations along with robot proprioception. Apart from the new task variants, we report the baseline data generation and agent performance statistics present in the MimicGen paper [11].

**Motion Planning.** In both the simulation and real-world tasks, we use TRAC-IK [80] for inverse kinematics, RRT-Connect [81] for joint-space motion planning, and Operational-Space Control (OSC) for task-space control [82]. In simulation, we check collisions during planning using the ground-truth obstacle collision geometries. In the real world, because collision geometries are not known, we use point-cloud-based collision checking using the segmented point cloud.

## 6 Experiments

### 6.1 SkillGen Features

**SkillGen improves data generation rates over MimicGen substantially.** MimicGen uses replay-based data generation for the entire trajectory, while SkillGen only uses replay for short skill segments, deferring larger transit motions to a motion planner. This results in substantially higher data generation success rates compared to MimicGen (average 75.4% vs. 40.7%, see Appendix F), especially when the reset distribution is large compared to the source demonstrations. Some compelling examples include Square $D_2$ (87.7% vs. 31.8%), Threading $D_2$ (74.3% vs. 21.6%), Three Piece Assembly $D_2$ (69.3% vs. 31.3%), and Coffee $D_2$ (70.0% vs. 27.7%).

**SkillGen data collection is robust to large object rearrangements and clutter.** In Coffee Prep $D_2$, the drawer containing the coffee pod and the mug are on opposite ends of the table compared to $D_0$ (source demos), and MimicGen is unable to collect any demonstrations while SkillGen achieves 59.9% data generation success. Additionally, in the Clutter variants of Square and Coffee (Ap-

| Task Variant | Src | MG | HSP-T | HSP-C | HSP-R |
|---|---|---|---|---|---|
| Square $D_0$ | 50.0 | 90.7 | **100.0** | **100.0** | 94.0 |
| Square $D_1$ | - | 73.3 | **100.0** | 98.0 | 62.0 |
| Square $D_2$ | - | 49.3 | **94.0** | **94.0** | 52.0 |
| Threading $D_0$ | 64.0 | 98.0 | **100.0** | 92.0 | 94.0 |
| Threading $D_1$ | - | 60.7 | **72.0** | 66.0 | 60.0 |
| Threading $D_2$ | - | 38.0 | **62.0** | 50.0 | **62.0** |
| Piece Assembly $D_0$ | 28.0 | 82.0 | **96.0** | 80.0 | 86.0 |
| Piece Assembly $D_1$ | - | 62.7 | **88.0** | 78.0 | 78.0 |
| Piece Assembly $D_2$ | - | 13.3 | **84.0** | 74.0 | 50.0 |
| Coffee $D_0$ | 100.0 | 100.0 | 100.0 | 100.0 | 100.0 |
| Coffee $D_1$ | - | 90.7 | 100.0 | 100.0 | 100.0 |
| Coffee $D_2$ | - | 77.3 | 94.0 | 100.0 | 98.0 |
| Nut Assembly $D_0$ | 22.0 | 60.0 | **100.0** | 92.0 | 94.0 |
| Nut Assembly $D_1$ | - | 16.0 | 72.0 | **78.0** | 20.0 |
| Nut Assembly $D_2$ | - | 12.0 | **54.0** | 50.0 | 24.0 |
| Coffee Prep $D_0$ | 2.0 | **97.3** | 92.0 | 92.0 | 84.0 |
| Coffee Prep $D_1$ | - | 42.0 | 54.0 | **74.0** | 64.0 |
| Coffee Prep $D_2$ | - | 0.0 | 80.0 | 74.0 | **84.0** |
| **Average** | - | 59.1 | **85.7** | 82.9 | 72.6 |

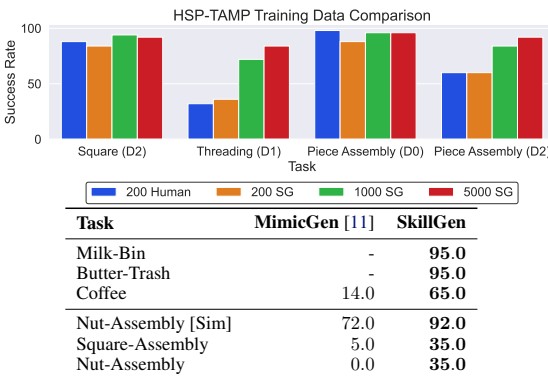

| Task | MimicGen [11] | SkillGen |
|---|---|---|
| Milk-Bin | - | **95.0** |
| Butter-Trash | - | **95.0** |
| Coffee | 14.0 | **65.0** |
| Nut-Assembly [Sim] | 72.0 | **92.0** |
| Square-Assembly | 5.0 | **35.0** |
| Nut-Assembly | 0.0 | **35.0** |

Figure 4: (*left*) **Agent Performance on SkillGen Datasets.** Success rates of agents trained on source demonstrations (with HSP-TAMP), MimicGen [11] data (with BC-RNN [1]), and SkillGen data (with all HSP variants). SkillGen data greatly improves agent performance on $D_0$ compared to the source data, and SkillGen agents substantially outperform MimicGen agents, especially on more challenging task variants. (*upper right*) **Training Data Comparison.** HSP-TAMP agent performance is comparable on 200 SkillGen demos and 200 human demos, despite SkillGen using just 10 human demos for generation. Generating more SkillGen demonstrations can result in significant performance improvement (also see Appendix E). (*lower right*) **Real-World Manipulation Results.** HSP-Class agents trained on SkillGen data generated in the real world are proficient, and substantially outperform using MimicGen data. They can also be transferred zero-shot from sim-to-real.

pendix D), a large object is placed randomly on the table. SkillGen achieves data generation rates from 49.0% to 72.0% while MimicGen only achieves 4.0% to 16.5%.

**SkillGen greatly improves agent performance on the source task.** Comparing HSP-TAMP agents trained on the source data vs. on SkillGen data on $D_0$, we see dramatic improvement (Fig. 4) – some examples include Three Piece Assembly (28% to 96%) and Nut Assembly (22% to 100%).

**SkillGen produces more proficient agents through its use of hybrid control.** Averaged across all tasks, HSP-TAMP, HSP-Class, and HSP-Reg achieve 85.7%, 82.9%, and 72.6% success rates respectively, compared to 59.1% for agents trained on MimicGen data (Fig. 4). Furthermore, HSP-Class and HSP-Reg make fewer assumptions than HSP-TAMP (see Sec. 4) while retaining the benefits of hybrid control. On Nut Assembly $D_1$ and $D_2$, HSP agents trained on SkillGen data outperform agents trained on MimicGen data by up to 62%, and SkillGen is able to train proficient agents (74% to 84%) on Coffee Prep $D_2$, while MimicGen fails to generate data for this variant (Fig. 4).

**SkillGen effectively adapts demonstrations across robots.** We use source demonstrations collected on the Panda arm and generate demonstrations for the Sawyer arm. As shown in Appendix N, data generation rates and policy performances are much higher for SkillGen than MimicGen.

## 6.2 SkillGen Analysis

**Can agent performance on SkillGen data match agent performance on an equal amount of human demonstrations?** We collected 200 demonstrations with the HITL-TAMP system [13] on each of 4 tasks and compared HSP-TAMP agent performance (the same method from HITL-TAMP) on the 200 human demos vs. 200 SkillGen demos (Fig. 4) generated from just 10 HITL-TAMP demos (which took less than 4 minutes per task to collect, compared to 37-71 minutes). Performance is comparable across all 4 tasks – 10% is the largest deviation, showing that SkillGen generated data is as effective as an equal number of human demos but only requires a small fraction of the effort.

**Does agent performance improve by generating more demonstrations?** We compared the performance of the different HSP algorithms on 200, 1000, and 5000 SkillGen demonstrations across the same 4 tasks from above – the results are presented in Fig. 4 (HSP-TAMP), and Appendix E (HSP-Class, HSP-Reg). All tasks and methods receive a significant increase from 200 to 1000 demos, and some tasks benefit strongly from 1000 to 5000 demos, notably Square $D_2$ (52% to 72% on HSP-Reg) and Threading $D_1$ (60% to 76% on HSP-Reg).

**How does performance compare between the different hybrid control learning algorithms?**
Average task performance between HSP-TAMP and HSP-Class is similar (85.7% vs. 82.9%), and
only slightly lower for HSP-Reg (72.2%) despite HSP-Class and HSP-Reg making much fewer
assumptions (Fig. 4). HSP-Reg results could improve with more SkillGen data (Appendix E).

### 6.3 Real World Evaluation

We first demonstrate that SkillGen data generation can be deployed in the real-world and the data
enables proficient policies to be learned. Next, we transfer agents trained in simulation with SkillGen
zero-shot to the real-world on a long-horizon task, demonstrating that combining SkillGen with
more sophisticated sim-to-real approaches is a promising method for robots to acquire real-world
manipulation capabilities with minimal human effort. Results are summarized in Fig. 4 (lower right).

**Setup.** We use a Panda robot arm, a front-view RealSense D415 camera, and a wrist-view RealSense
D435 camera. Pose estimates are obtained using FoundationPose [83]. Agents use proprioception
and 120x160 camera images (except for sim-to-real agents) and are evaluated over 20 rollouts.

**SkillGen Data Generation and Policy Learning in the Real World.** We collect 3 source demon-
strations with HITL-TAMP teleoperation on each of our tasks (Pick-Place-Milk, Cleanup-Butter-
Trash, and Coffee), use SkillGen to generate 100 demonstrations, and train HSP-Class agents on
the generated data (Appendix J has full details). These agents obtain near-perfect success rates
on the Pick-Place-Milk and Cleanup-Butter-Trash tasks despite large amounts of spatial variation.
HSP-Class also obtains 65% on the challenging Coffee task, while the BC-RNN agent trained on
MimicGen data from [11] could only obtain 14%. This result is comparable with the 74% reported
in HITL-TAMP [13] for an HSP-TAMP agent trained with 100 HITL-TAMP demos. We note the
lower human effort (3 human demos vs. 100), that our Coffee task is more challenging (requires
agent to learn to grasp the pod, unlike [13]) and our HSP-Class agent makes fewer assumptions.

**Zero-Shot Sim-to-Real Deployment of SkillGen Policies.** We designed a simulation task (Nut-
Assembly [Sim]) that mirrors our real-world "Nut Assembly" task, where the robot must grasp a
square and round nut and fit them onto corresponding square and round pegs. We train agents
in simulation by collecting 1 source demo (with HITL-TAMP for SkillGen and with conventional
teleoperation for MimicGen), generate 1000 demonstrations with SkillGen and MimicGen, and sub-
sequently train an HSP-Class agent and a MimicGen (BC-RNN) agent (see Fig. 4, lower right). This
task is challenging even in simulation, as the trained simulation agents are imperfect (HSP-Class:
92%, MimicGen: 72%). When deployed on the real-world task, the MimicGen agent manages to
solve the first insertion task (Square-Assembly) with 5% success rate, but never solves the full task
while the HSP-Class agent is able to achieve 35% success rate. This shows the value of SkillGen's
hybrid control paradigm in aiding sim-to-real transfer through decomposing tasks into a sequence
of local behaviors that are more likely to transfer [84]. More details and discussion in Appendix K.

## 7 Limitations

SkillGen requires knowledge of a fixed sequence of skills that can complete a task. It assumes that
object poses can be observed at the start of each skill segment during data generation. SkillGen was
demonstrated on quasi-static tasks involving rigid objects. SkillGen produces the best results when
using source human demonstrations collected with the HITL-TAMP system – improving results
with conventional teleoperation is left for future work. In the sim-to-real experiment, the agents had
limited observability. Namely, agents only observe changes in proprioception, as no pose tracking
or visual observations are used during execution. See Appendix C for full discussion.

## 8 Conclusion

We introduced SkillGen, a data generation system that synthesizes large datasets by adapting select
skill segments from a handful of human demonstrations, and a Hybrid Skill Policy (HSP) learning
framework to learn from the generated datasets by enabling closed-loop skills to be sequenced using
a motion planner. We showed that SkillGen improves over a state-of-the-art data generation sys-
tem, in both data generation capability and the ability to learn proficient agents from the data. We
demonstrated SkillGen on real-world manipulation tasks, including zero-shot sim-to-real transfer.

**Acknowledgments**

This work was made possible due to the help and support of Yashraj Narang (assistance with CAD asset design and helpful discussions), Michael Silverman, Kenneth MacLean, and Sandeep Desai (robot hardware design and support), and Ravinder Singh (IT support).

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

# Appendix

## A    Overview

The Appendix contains the following content.

- **FAQ** (Appendix B): answers to some common questions
- **Limitations** (Appendix C): more thorough list and discussion of SkillGen limitations
- **Analysis on Challenging Data Generation Scenarios** (Appendix D): more results and discussion on challenging data generation scenarios addressed by SkillGen
- **Dataset Scaling Law Analysis** (Appendix E): full set of results for generating larger datasets with SkillGen
- **Data Generation Success Rates** (Appendix F): data generation success rates for SkillGen datasets
- **Data Generation Details** (Appendix G): more details on how SkillGen generates data
- **Policy Learning Details** (Appendix H): more details on how policies were trained from SkillGen datasets
- **Planning Details** (Appendix I): more details on the planners used in this work
- **Tasks and Task Variants** (Appendix J): detailed descriptions of tasks and task variants used to evaluate SkillGen
- **Sim-to-Real Experiment** (Appendix K): details on sim-to-real experiments
- **Results with Conventional Teleoperation Source Demonstrations** (Appendix L): Skill-Gen performance on conventional teleoperation source demos
- **Ablations** (Appendix M): ablations of certain data generation and policy learning components
- **Robot Transfer** (Appendix N): SkillGen applied to generate data and train policies across robot arms
- **Algorithm Pseudocode** (Appendix O): pseudocode for SkillGen data generation and policy deployment
- **Comparison with HITL-TAMP [13]** (Appendix P): more discussion on how SkillGen compares with HITL-TAMP
- **Discussion on HSP-Reg Results** (Appendix Q): more discussion on the gap between HSP-Reg and other methods and additional promising results
- **Skill Segments and Annotations** (Appendix R): more commentary on skill segments and how they can be annotated in the source data
- **Comparison with Replay-Noise Baseline** (Appendix S): comparison of SkillGen to a baseline that replays the source demonstrations with noise added to the actions
- **Results Across Multiple Seeds** (Appendix T): policy learning results across multiple seeds

# B  FAQ

1. **What are some limitations of SkillGen?**

   See Appendix C.

2. **Why might a data generation attempt in a failure?**

   The transformed human segments (skill segments) during data generation (Sec. 4.4) might result in poses that are difficult or impossible for the motion planner or task-space controller to reach. Small errors can also accumulate during open-loop replay of the skill segments, causing failures during high-precision motions such as insertion. Despite the potential for these failures, proficient agents can be trained from SkillGen datasets.

3. **Are there concrete examples of situations where SkillGen succeeds in generating data but MimicGen fails?**

   See Appendix D.

4. **Is SkillGen compatible with normal teleoperation systems or do I have to use HITL-TAMP?**

   Yes, SkillGen is compatible with normal teleoperation systems – see Appendix L for results and discussion.

5. **What are the assumptions made by each HSP policy learning method?**

   HSP-Reg makes no additional assumptions compared to standard Behavioral Cloning methods. HSP-Class makes similar assumptions to those made during data generation – namely that the sequence of relevant objects that the robot must interact with for a task are known, and we are able to observe or estimate object poses prior to robot interaction (Sec. 3, A2 and A3). Importantly, this does not require full object pose tracking. HSP-TAMP [13] makes the most assumptions. It assumes access to a TAMP system that knows where to move the robot before initiating the learned skill policy and when to terminate the learned skill policy.

6. **There is a small but significant performance gap between HSP-Reg, and the other HSP methods. Does that mean that policies must use privileged information to get the benefits of the HSP skill formulation?**

   The results are close between HSP-Reg and the other methods in many cases (Fig. 4, average success rate only lower by 10% to 13%) despite making much fewer assumptions (see FAQ (5) above). However, there are some easy ways to improve performance further (discussion in Appendix Q), including generating more data (Appendix E). Moreover, HSP-Reg might be the only method appropriate for tasks in which, for example, the objects vary.

7. **Is it necessary for SkillGen data generation rates to be high for policies trained on the generated demo to perform well? If not, why is it beneficial to have higher data generation rates?**

   There isn't a strict correlation between data generation success rate and trained policy success rate. In many cases, data generation success rates can be very low, especially when using initiation augmentation (Appendix F), compared to the resulting policy success rates. However, higher data generation rates can be beneficial for generating datasets more quickly (in terms of wall clock time), since it will take less time to reach a target amount of data. Even when data generation rates are low, SkillGen can leverage parallelization during data generation to generate data faster (Appendix G.4). Finally, a higher data generation rate can imply better coverage of the task reset distribution in the generated data, but a low data generation rate does not necessarily mean the task reset distribution is not covered well.

8. **Can SkillGen be used to generate data for different robot arms, like MimicGen?**

   Yes, see Appendix N for results.

9. **Explain how SkillGen was used to generate over 24K demonstrations across 18 task variants in simulation from just 60 human demonstrations.**

   We generated 1000 SkillGen demos for each of the 18 task variants in Fig. 4 and an additional 6 more datasets (1000 demos each) with a different robot arm (Appendix N), using just 10 source human demos collected on the 6 simulation tasks. We do not include the

dataset scaling law experiments (Appendix E), the datasets generated with initiation augmentation, and the datasets generated in the real world, which would increase the total substantially.

10. **How does SkillGen compare to a baseline that replays existing source demonstrations with noise?**

   We present this comparison in Appendix S and show that SkillGen outperforms this baseline, especially when the reset distribution is large. Furthermore, this baseline cannot generate data for new reset distributions, unlike SkillGen.

# C  Limitations

We discuss limitations of SkillGen that can inform future work, extending Section 7.

1. **Given sequence of skill segments during data generation.** During data generation, the sequence of skill segments (relevant objects that must be manipulated by the robot during each skill) must be provided.

2. **Object pose estimates during data generation.** During data generation, SkillGen assumes access to the object pose at the start of each skill segment, either by direct observation (simulation) or estimation (real world).

3. **Quasi-static tasks with rigid objects.** This paper applies SkillGen to primarily quasi-static tasks with rigid objects.

4. **Better performance when using source human data from HITL-TAMP [13] than from conventional teleoperation systems.** SkillGen obtains better results when using human demonstrations collected with HITL-TAMP than with conventional teleoperation systems (Appendix L). Investigating how more consistent human annotations can reduce this gap is future work.

5. **Limited agent observability and action space for sim-to-real experiments.** Agents used in the sim-to-real experiments only observe changes in robot proprioception, as no pose tracking or visual observations are used during execution. The agent also receives object poses at the start of each episode, but these are never updated. The action space is restricted to position-only control (no rotation). These design choices were made to maximize the possibility of transfer without the need for addressing the gap in perception between simulation and the real world, and without the need for extensive robot controller tuning between simulation and the real world. See Appendix K for more details and discussion.

# D  Analysis on Challenging Data Generation Scenarios

In this section, we discuss some challenging data generation scenarios where SkillGen is able to generate data, while MimicGen struggles. We first review some limitations of MimicGen, and then we discuss different data generation scenarios.

## D.1  MimicGen Limitations

**Susceptibility to scene collisions.** MimicGen uses a naive linear interpolation scheme during data generation to connect the end of one transformed object-centric human segment to another one. This approach is not aware of scene geometry, which can result in data generation failures due to collisions between the robot and other objects in the scene. By contrast, SkillGen transit and transfer motions between skill segments are carried out via motion planning.

**Tradeoff between Data Generation Quality and Policy Learning Proficiency.** The use of naive linear interpolation also impacts learning ability. Longer in time (not space) interpolation segments have been shown to be harmful to policies trained from MimicGen data [11], which motivates the use of short interpolation segments with a small number of intermediate waypoints. However, this can lower the data generation success rate, since the end-effector controller might not be capable of accurately tracking waypoints that are far apart, and this also can be unsafe for real-world deployment. Consequently, MimicGen has a fundamental tradeoff with respect to interpolation segments. On one hand, shorter segments are better for policy learning but can result in lower data generation success rates and be unsafe for real-world deployment. On the other, longer segments are more suitable for real-world deployment and for ensuring better data generation throughput but make policy learning more difficult. By contrast, SkillGen has no such tradeoff.

## D.2  Challenging Data Generation Scenarios

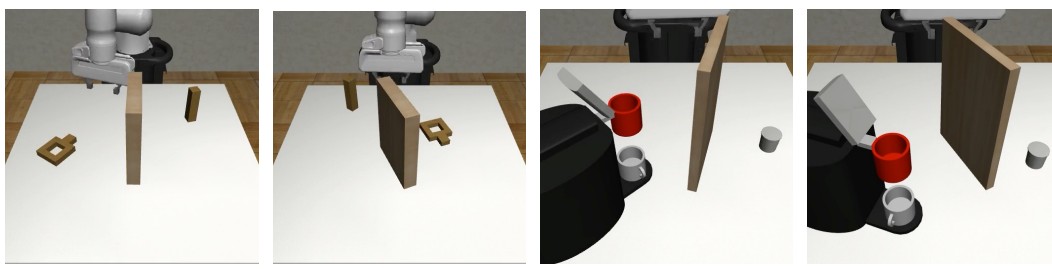

Figure D.1: **Example Configurations for Clutter Tasks.** Example configurations from the clutter task variants of Square and Coffee.

**Presence of Clutter.** SkillGen successfully generates data for scenes with large obstacles, unlike MimicGen. We develop variants of the Square and Coffee tasks that have a large obstruction placed in the workspace (Fig. D.1). The reset distributions for these tasks are identical to their clutter-free counterparts described in Appendix J except for the presence of the obstruction, which has its own reset distribution, and is placed randomly near the center of the workspace. We use the same source demonstrations as before (collected on the clutter-free $D_0$ variants of these tasks) and perform 200 data generation attempts with both SkillGen and MimicGen. The data generation success rates are presented in Table D.1. We see that SkillGen substantially outperforms MimicGen by margins as large as 58.5%.

| Task Variant | MimicGen [11] | SkillGen |
|---|---|---|
| Square ($D_1$, Clutter) | 4.0 | **62.5** |
| Square ($D_2$, Clutter) | 14.5 | **72.0** |
| Coffee ($D_0$, Clutter) | 16.5 | **49.0** |
| Coffee ($D_1$, Clutter) | 14.0 | **55.0** |

Table D.1: **Data Generation Rates for Environments with Clutter.** SkillGen is able to generate data for environments with clutter much more effectively than MimicGen.

**Large Scene Variations from Source Demos.** SkillGen excels at generating data even when there are substantial deviations from where objects were located in the source human demonstrations unlike MimicGen, which suffers from having to use short linear interpolation segments during generation. For example, MimicGen is unable to produce any data on Coffee Prep $D_2$, due to the mug and drawer being on opposite sides of the table compared to the source demos ($D_0$) (see Fig. D.2), while SkillGen can generate data and train proficient agents on $D_2$ (Fig. 4). SkillGen also enjoys large gains over MimicGen for data generation rates, especially on $D_2$ task variants, which vary substantially from $D_0$, where source data was collected. This can be seen in Table F.1 (Appendix F).

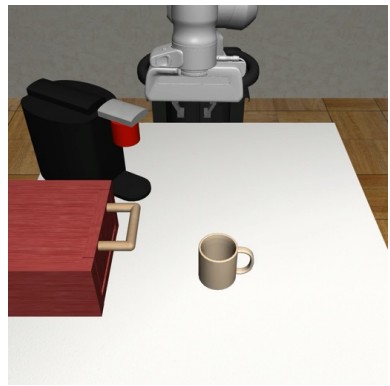 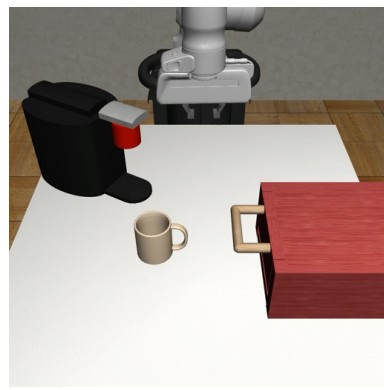

Figure D.2: **Coffee Prep $D_0$ and $D_2$.** Example configurations for two task variants of Coffee Prep. Source demonstrations were collected on $D_0$. MimicGen is unable to generate data on $D_2$ due to the drawer and mug being on opposite ends of the table compared to the source demos, while SkillGen successfully generates data and trains proficient policies for $D_2$.

**Safe and Proficient Real World Deployment.** SkillGen is able to obtain proficient policies in the real world, as shown in Sec. 6.3, unlike MimicGen. MimicGen had to use longer interpolation segments in the real world, to enforce safety during execution, which made policy learning results suffer (as discussed above).

# E    Dataset Scaling Law Analysis

We present results for using different amounts of SkillGen data for policy training, to see how policy success rate scales with amount of data. We present results in Table E.1 (for HSP-TAMP), Table E.2 (for HSP-Class) and Table E.3 (for HSP-Reg). HSP-Reg uses SkillGen with initiation augmentation (Sec. 4.5). All tasks and methods receive a significant increase from 200 to 1000 demos, and some tasks benefit strongly from 1000 to 5000 demos, notably Square $D_2$ (52% to 72% on HSP-Reg) and Threading $D_1$ (60% to 76% on HSP-Reg).

| Task Variant | Human 200 | SkillGen 200 | SkillGen 1000 | SkillGen 5000 |
|---|---|---|---|---|
| Square ($D_2$) | 88.0 | 84.0 | **94.0** | 92.0 |
| Threading ($D_1$) | 32.0 | 36.0 | 72.0 | **84.0** |
| Piece Assembly ($D_0$) | **98.0** | 88.0 | 96.0 | 96.0 |
| Piece Assembly ($D_2$) | 60.0 | 60.0 | 84.0 | **92.0** |

Table E.1: **Policy Training Dataset Comparison with HSP-TAMP [13].** Table of results corresponding to the comparison in Fig. 4 (upper right). HSC-TAMP agent performance is comparable on 200 SkillGen demos and 200 human demos, despite SkillGen using just 10 human demos for generation. Generating more SkillGen demonstrations can result in significant performance improvement.

| Task Variant | SkillGen 200 | SkillGen 1000 | SkillGen 5000 |
|---|---|---|---|
| Square ($D_2$) | 74.0 | 94.0 | **96.0** |
| Threading ($D_1$) | 34.0 | 66.0 | **80.0** |
| Piece Assembly ($D_0$) | 72.0 | 80.0 | **86.0** |
| Piece Assembly ($D_2$) | 44.0 | 74.0 | **78.0** |

Table E.2: **Policy Training Dataset Comparison with HSP-Class.** Generating more SkillGen demonstrations can result in modest performance improvement.

| Task Variant | SkillGen 200 | SkillGen 1000 | SkillGen 5000 |
|---|---|---|---|
| Square ($D_2$) | 4.0 | 52.0 | **72.0** |
| Threading ($D_1$) | 14.0 | 60.0 | **76.0** |
| Piece Assembly ($D_0$) | 68.0 | **86.0** | 82.0 |
| Piece Assembly ($D_2$) | 2.0 | 50.0 | **62.0** |

Table E.3: **Policy Training Dataset Comparison with HSP-Reg.** Generating more SkillGen demonstrations can result in substantial performance improvement for certain tasks.

# F  Data Generation Success Rates

We present data generation rates for the datasets used in our experiments (Table F.1 for simulation tasks and Table F.2 for real-world tasks and the sim-to-real task). In most cases, SkillGen achieves higher data generation rates than MimicGen. One notable exception is when using initiation augmentation (Sec. 4.5) – success rates are much lower in this case. However, this is due to the aggressive noise distribution applied to motion planner targets during the generation process. See Appendix G.3 for more discussion.

| Task Variant | MimicGen [11] | SkillGen | SkillGen (+IA) |
|---|---|---|---|
| Square ($D_0$) | 73.7 | **99.8** | 30.7 |
| Square ($D_1$) | 48.9 | **91.5** | 34.3 |
| Square ($D_2$) | 31.8 | **87.7** | 27.5 |
| Threading ($D_0$) | 51.0 | **76.2** | 35.0 |
| Threading ($D_1$) | 39.2 | **66.4** | 27.2 |
| Threading ($D_2$) | 21.6 | **74.3** | 24.9 |
| Piece Assembly ($D_0$) | 35.6 | **82.5** | 5.1 |
| Piece Assembly ($D_1$) | 35.5 | **72.7** | 4.7 |
| Piece Assembly ($D_2$) | 31.3 | **69.3** | 4.6 |
| Coffee ($D_0$) | **78.2** | 73.3 | 9.3 |
| Coffee ($D_1$) | 63.5 | **73.6** | 9.1 |
| Coffee ($D_2$) | 27.7 | **70.0** | 8.5 |
| Nut Assembly ($D_0$) | 53.0 | **98.6** | 15.2 |
| Nut Assembly ($D_1$) | 30.0 | **91.7** | 15.1 |
| Nut Assembly ($D_2$) | 22.8 | **69.1** | 10.6 |
| Coffee Prep ($D_0$) | 53.2 | **64.6** | 1.4 |
| Coffee Prep ($D_1$) | **36.1** | **36.8** | 0.7 |
| Coffee Prep ($D_2$) | 0.0 | **59.9** | 0.6 |
| Average | 40.7 | **75.4** | 14.7 |

Table F.1: **Data Generation Rates for Simulation Environments.** SkillGen improves data generation rates over MimicGen substantially. When using initiation augmentation (+IA), data generation rates are much lower, due to the aggressive noise distribution applied to motion planner targets.

| Task | MimicGen [11] | SkillGen |
|---|---|---|
| Pick-Place-Milk | - | 100.0 |
| Cleanup-Butter-Trash | - | 95.0 |
| Coffee | 52.0 | 73.0 |
| Nut-Assembly [Sim] | 72.6 | 94.8 |

Table F.2: **Data Generation Results on Real World Manipulation Tasks and Sim-to-Real Tasks.** SkillGen has high data generation throughput even in the real world, and compares favorably to MimicGen. The bottom part of the table shows the data generation rate in simulation for the task used for sim-to-real transfer.

# G  Data Generation Details

In this section, we provide more details on SkillGen data generation. We first describe how reference skill segments are selected, and how they are transformed and executed. We next describe how initiation augmentation can be used to produce more robust closed-loop agents. Finally, we describe how we leveraged parallelization to generate large datasets efficiently with reasonable wall clock times, even when data generation rates were low.

## G.1  Reference Skill Segment Selection

During a data generation attempt, SkillGen adapts existing skill segments to the new scene and executes them sequentially with motion segments (Sec. 4.4). To generate a new skill segment (for skill index $i$ for a task), SkillGen requires a reference skill segment $\tau_{is}$ to be selected from the source demonstrations $D_{\mathrm{src}}$. Since the skill index should match between the source demonstrations and the current skill segment that must be generated, this problem reduces to selecting a source demonstration index $j \in \{1, 2, ..., N\}$. In our experiments, we sample this index randomly for the first skill segment, and then leave it fixed for the rest of the episode. However, more sophisticated selection methods could be used to select a different source demonstration index for each skill index if desired.

## G.2  Skill Segment Execution and Action Noise

During a data generation attempt, after an existing skill segment is selected and transformed to obtain a new sequence of end-effector pose actions $\tau'_{is} = (T_W^{A'_0}, ..., T_W^{A'_K})$ (Sec. 4.4), this sequence of actions is executed one by one. However, we found it beneficial to apply additive noise to the pose actions. As in MimicGen, we convert each absolute pose action to a normalized delta pose action (using the current robot end effector pose) and add Gaussian noise $\mathcal{N}(0, 1)$ with magnitude $\sigma$ in each dimension, where $\sigma = 0.05$. Note that the gripper actuation actions are copied as-is from the source skill segment, and no noise is added. These modified normalized delta pose actions are then executed, and stored in the generated dataset.

## G.3  Initiation Augmentation

As described in Sec. 4.5, SkillGen has the option of adding noise to the skill initiation states $T_W^{E_0}$, producing new initiation states $T_W^{E'_0}$, to broaden the support of the initiation set and allow the trained closed-loop skill policies to be more robust to incorrect initiation pose predictions. We found this to be very helpful for HSP-Reg agents, which must directly predict initiation poses via regression. Consequently, all of our HSP-Reg agents are trained on datasets with initiation augmentation, unless otherwise noted.

For datasets generated with initiation augmentation, we add uniform translation noise to the target position for each initiation state $\mathcal{U}[-t, t]$, where $t$ is the position noise scale. We also modify the target rotation, by sampling a random rotation axis (random vector on 3D unit sphere), sampling a random angle $\phi \sim \mathcal{U}[0, r]$, converting the new sampled axis-angle rotation to a rotation matrix, and multiplying the target rotation by this rotation matrix. The motion planner will attempt to reach the new target pose, and then we will subsequently plan and execute a *recovery segment* consisting of a sequence of pose actions that moves from new pose $T_W^{E'_0}$ to the original pose $T_W^{E_0}$. The recovery segment is added to the transformed skill segment, and is part of the dataset used to train the closed-loop agent.

In our experiments, we chose $t = 0.08$ meters and $r = 80$ degrees. We note that this is a very wide and aggressive pose randomization distribution, and that a large portion of sampled poses will be unreachable by the motion planner, due to the pose being in collision with the scene. This is why the data generation rates for the initiation augmentation datasets are significantly lower (Appendix F). This could be addressed with more intelligent sampling mechanisms, but we leave this for future work. Instead, opted to leverage parallelization during data generation to efficiently generate large datasets in a reasonable amount of wall clock time (described below).

### G.4 Efficient Data Generation with Parallelization

Datasets generated with initiation augmentation can have low data generation rates due to the broad noise distribution and rejection sampling process used. To mitigate this, we parallelized data collection across a large number of cpu processes. The SkillGen data generation process is easily amenable to this type of parallelization.

### G.5 Hardware

Data generation runs were batched together and run simultaneously (on a compute cluster) on 8-GPU nodes consisting of 8 NVIDIA Volta V100 GPUs, 64 CPUs, and 400GB of memory. Real robot experiments were run on a machine with an NVIDIA GeForce RTX 3090 GPU, 36 CPUs, 32GB of memory, and 1 TB of storage.

# H Policy Learning Details

Here, we describe how policies are trained with SkillGen data. All policies trained on MimicGen data are trained with BC-RNN [1] using the same hyperparameters as in MimicGen [11].

## H.1 Observation Spaces

Every network used camera observations, consisting of a front-view camera and a wrist-view camera, and proprioception consisting of end effector poses and gripper finger positions unless otherwise mentioned. The simulation tasks used an image resolution of 84x84 and the real-world tasks used an image resolution of 120x160. All networks taking image inputs utilize pixel shift randomization [1, 47–50] to shift image pixels by up to 10% of each dimension randomly on each forward pass.

## H.2 Policy Evaluation

Unless otherwise mentioned, policies are evaluated using 50 rollouts per checkpoint during training. The best-performing policy success rate is reported for each training run [1].

## H.3 Training Procedures and Hyperparameters

We outline how each network used by the HSP algorithms described in Sec. 4.6 is trained. All networks are trained with the Adam optimizer [85] with a learning rate of 1e-4. Only one network of each type is used across all skill segments (e.g. we do not train separate networks per skill).

**Policy Network ($\pi_\theta$) (HSP-Reg, HSP-Class, HSP-TAMP):** The policy network is trained with BC-RNN using robomimic [1] using the same default network structure and hyperparameters from their study. This matches the settings used for training policies in MimicGen [11].

**Termination Classifier ($\mathcal{T}_\theta$) (HSP-Reg, HSP-Class):** This is a binary classification network $\mathcal{T}_\theta : \mathcal{O} \rightarrow \{0, 1\}$ that is trained to predict when the skill policy should be running. The network architecture uses the same observation encoder structure (with different learned weights) as the policy network – each image is encoded using a ResNet-18 network [86] followed by a spatial-softmax layer [87], and these outputs are concatenated directly with the other non-image observations. This is then fed to an MLP with 2 hidden layers of size 1014, which outputs 2 logits. The network is trained with a standard multi-class classification Cross-Entropy loss. Labels to train this network are easily obtained from the SkillGen dataset, as each observation-action pair $(o, a)$ is labeled with whether it was collected while the motion planner was running or not. We additionally apply data augmentation, and flip the labels on the last 50% of each motion planner segment. This is useful to ensure the termination classifier does not erroneously predict that the policy should terminate at the start of the skill segment. During agent rollouts, we additionally only accept a valid termination prediction when termination has been predicted 5 times – we found this to be a simple mechanism to prevent early termination prediction.

**Initiation Regression Network ($\mathcal{I}_\theta$) (HSP-Reg):** This is a network $\mathcal{I}_\theta : \mathcal{O} \rightarrow SE(3)$ that directly predicts an end effector pose corresponding to the initiation condition for the next skill. The architecture is the same as the termination classifier, except for the last layer, which directly predicts a position (3-dim) and a rotation (6-dim rotation representation from Zhou et al [88]). To allow for multimodal predictions, we use a Gaussian Mixture Model (GMM) head, using the same hyperparameters as the BC-RNN-GMM model from robomimic [1]. The position and rotation targets to train the network come from the SkillGen dataset, and correspond to the targets that were sent to the motion planner during data generation. These targets are normalized to lie in $[-1, +1]$, using the same procedure from Chi et al. [89]. During agent rollouts, this network directly samples a target pose for the motion planner to reach.

**Initiation Classifier ($\mathcal{I}_\theta$) (HSP-Class):** This is a classification network $\mathcal{I}_\theta : \mathcal{O} \rightarrow \{1, 2, ..., N_{\text{src}}\}$ that frames skill initiation condition prediction as a classification problem over initiation states in the source dataset $\mathcal{D}_{\text{src}}$. The architecture is the same exact network (with shared weights) as the termination classifier described above ($\mathcal{T}_\theta$) – there is simply an extra classification head added to the output of the network. It is trained to predict the source demonstration in $\mathcal{D}_{\text{src}}$ that spawned the generated demonstration in $\mathcal{D}$ using a standard multi-class classification Cross-Entropy loss. During

agent rollouts, after predicting a source demonstration label, the corresponding initiation state in the source demonstration is adapted to the current state using the adaptation procedure from Sec. 4.4 to obtain a target pose for the motion planner.

## H.4 Hardware

Policy learning runs each used a machine (on a compute cluster) with an NVIDIA Volta V100 GPU, 8 CPUs, and 50GB of memory. Real robot experiments were run on a machine with an NVIDIA GeForce RTX 3090 GPU, 36 CPUs, 32GB of memory, and 1 TB of storage.

# I  Planning Details

In this section, we provide additional details on the Motion Planner and the Task and Motion Planner used in our experiments, beyond those in Sec. 5. We used PyBullet [90] for collision checking during motion planning and TAMP. Within the HITL-TAMP system, we used PDDLStream [91] for task and motion planning.

For each motion planning query, we decompose planning into three phases. The first is a short *retreat* motion that moves the robot's end effector backward. The second is a transit or transfer motion that moves the robot a short distance in front of the query pose. The third is an *approach* motion that moves to the query pose. The retreat and approach motions move the robot out of and into contact respectively. During these short phases, we ignore expected collisions between the robot and any manipulated object along with collisions between manipulated objects and the environment. In our experiments, we used a retreat and approach distance of 5cm in the end effector's z axis.

Expanding on Section 6.3, in the real world, we assume manipulable object segmentation. While a number of choices on segmentation methods can be made [92, 93], we deploy a simple yet effective pipeline which works well in our setup. Specifically, we first perform RANSAC plane fitting to filter the table from the observed point cloud. Then, we use DBSCAN [94] to cluster the objects within the remaining point cloud. In settings where we have shape models for the objects, the object cloud segments are distinguished by comparing to their respective 3D models and we use FoundationPose [83] for object 6D pose estimation. Otherwise, we reconstruct collision volumes for the manipulable objects online by running marching cubes [95] on each segmented point cloud. For transfer motion planning, we detect whether a manipulable object is grasped by checking for contact between both the robot's fingers and the object in our planning model. While grasped, we assume the object is rigidly attached to the robot, modifying its collision geometry.

# J  Tasks and Task Variants

In this section, we provide detailed descriptions of all the tasks (Fig. J.1) and task variants. See the website (https://skillgen.github.io) for more visualizations. The action space for all tasks is a delta-pose action space (using an Operational Space Controller [82]) to control the arm), along with a gripper open/close command. Control occurs at 20 hz.

## J.1  Simulation Tasks

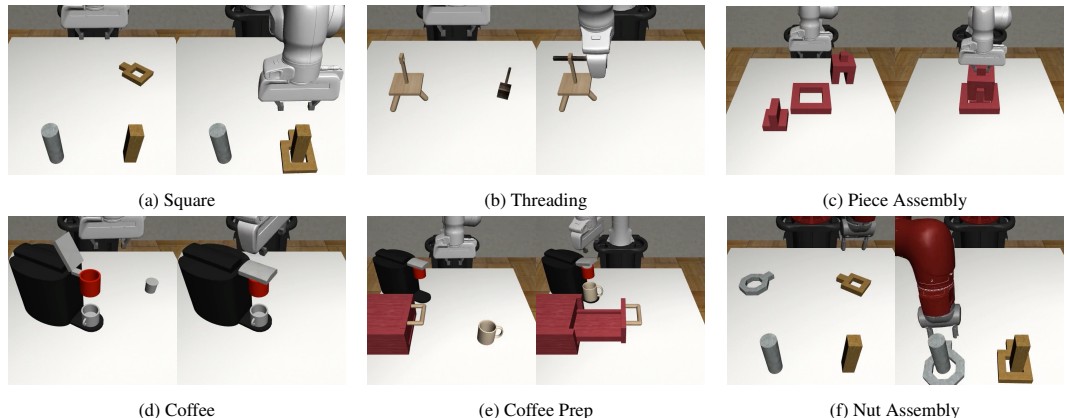

(a) Square  (b) Threading  (c) Piece Assembly

(d) Coffee  (e) Coffee Prep  (f) Nut Assembly

Figure J.1: **Simulation Tasks.** We deploy SkillGen on 6 simulation tasks (18 task variants). These tasks include fine-grained and long-horizon manipulation.

All tasks and task variants are taken from the MimicGen paper [11], with the exception of Nut Assembly ($D_1$, $D_2$) and Coffee Prep ($D_2$), which were newly implemented. For each task, we describe the goal, the task variants, and the skill segments.

- **Square.** The robot must pick a square nut and place it on a peg. ($D_0$) The peg never moves, and the nut is placed in small (0.005m x 0.115m) region with a random top-down rotation. ($D_1$) The peg and the nut are initialized in large regions, but the peg rotation is fixed. The peg is initialized in a 0.4m x 0.4m box and the nut is initialized in a 0.23m x 0.51m box. ($D_2$) The peg and the nut are initialized in larger regions (0.5m x 0.5m box of initialization for both) and the peg rotation also varies. There are 2 skill segments (grasp nut, place onto peg).

- **Threading.** The robot must pick a needle and thread it through a hole on a tripod. ($D_0$) The tripod is fixed, and the needle moves in a modest region (0.15m x 0.1m box with 60 degrees of top-down rotation variation). ($D_1$) The tripod and needle move in large regions on the left and right portions of the table respectively. The needle is initialized in a 0.25m x 0.1m box with 240 degrees of top-down rotation variation and the tripod is initialized in a 0.25m x 0.1m box with 120 degrees of top-down rotation variation. ($D_2$) The tripod and needle are initialized on the right and left respectively (reversed from $D_1$). The size of the regions is the same as $D_1$. There are 2 skill segments (grasp needle, thread into tripod).

- **Coffee**. The robot must pick a coffee pod, insert the pod into the coffee machine, and close the machine hinge. ($D_0$) The machine never moves, and the pod moves in a small (0.06m x 0.06m) box. ($D_1$) The machine and pod move in large regions on the left and right portions of the table respectively. The machine is initialized in a 0.1m x 0.1m box with 90 degrees of top-down rotation variation and the pod is initialized in a 0.25m x 0.13m box. ($D_2$) The machine and pod are initialized on the right and left respectively (reversed from $D_1$). The size of the regions is the same as $D_1$. There are 2 skill segments (grasp pod, insert-into and close machine).

- **Three Piece Assembly.** The robot must pick one piece, insert it into the base, then pick the second piece, and insert into the first piece to assemble a structure. ($D_0$) The base never moves, and both pieces move around base with fixed rotation in a 0.44m x 0.44m region. ($D_1$) All three pieces move in the workspace (0.44m x 0.44m region) with fixed rotation.

($D_2$) All three pieces can rotate (the base has 90 degrees of top-down rotation variation, and the two pieces have 180 degrees of top-down rotation variation). There are 4 skill segments (grasp piece 1, place into base, grasp piece 2, place into piece 2).

- **Nut Assembly.** Similar to Square, but the robot must place both a square nut and round nut onto two different pegs. ($D_0$) Each nut is initialized in a small box (0.005m x 0.115m region with a random top-down rotation). ($D_1$) The nuts are initialized in a large box (0.23m x 0.51m region) with random top-down rotation, and the pegs are initialized in a large box (0.4m x 0.4m) with a fixed rotation. ($D_2$) The nuts and pegs are initialized in a larger box (0.5m x 0.5m) with random top-down rotations. There are 4 skill segments (grasp each nut and place onto each peg).

- **Coffee Prep.** A more comprehensive version of Coffee — the robot must load a mug onto the coffee machine, open the machine, retrieve the coffee pod from the drawer and insert the pod into machine. ($D_0$) The mug moves in a modest (0.15m x 0.15m) region with fixed top-down rotation and the pod inside the drawer moves in a 0.06m x 0.08m region while the machine and drawer are fixed. ($D_1$) The mug is initialized in a larger region (0.35m x 0.2m box with random top-down rotation) and the machine also moves in a modest region (0.1m x 0.05m box with 60 degrees of top-down rotation variation). ($D_2$). Same task as $D_0$ but the drawer is placed on the right side of the table, and the mug is initialized on the left side of the table, instead of the right. There are 5 skill segments (grasp mug, place onto machine and open lid, open drawer, grasp pod, insert into machine and close lid).

## J.2 Real-World Tasks

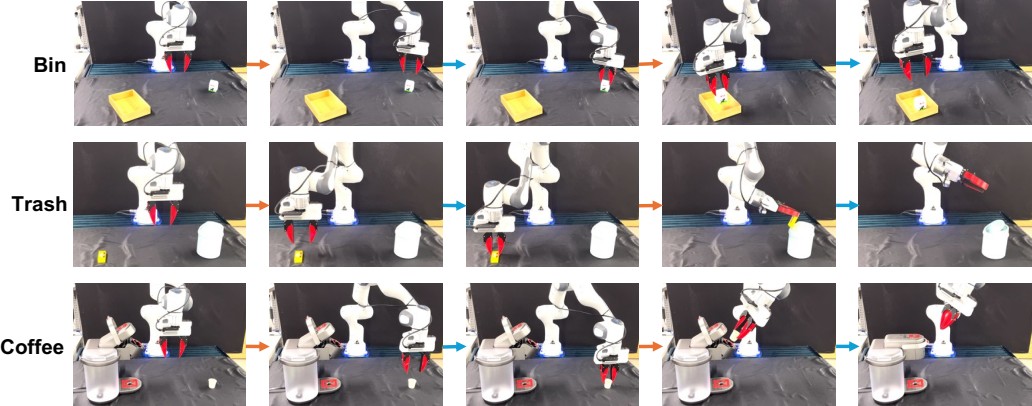

Figure J.2: **Real-World Task Executions.** The 1) initial state, 2) pick initiation state, 3) pick termination state, 4) placement or insertion initiation state, and 5) placement or insertion termination state for an example episode of the Milk-Bin, Butter-Trash, and Coffee tasks. The orange arrows indicate a transition facilitated by motion planning, and the blue arrows indicate a transition conducted by a learned skill policy.

Figure J.2 and Figure J.3 display example task executions and the initial state distributions respectively for the Pick-Place-Milk, Cleanup-Butter-Trash, and Coffee tasks introduced in Section 6.3. For each task, we describe the goal, the initialization regions for the objects, and the skill segments.

- **Pick-Place-Milk.** The robot must pick the milk and place it in the bin. The milk and bin objects are randomly placed anywhere on the table, with random orientations that are within +/-45 degrees of yaw from their nominal orientations. There are two skill segments: pick milk and place into bin.

- **Cleanup-Butter-Trash.** The robot must pick the butter and insert it into the trash can by pushing the trash can's lid. The butter and trash can are placed randomly on the left and right sides of the table respectively, with random orientations that are within +/-45 degrees of yaw from their nominal orientations. There are two skill segments: pick butter and insert into trash can.

- **Coffee.** The robot must pick the coffee pod, insert it into the coffee machine, and then close the machine's lid. The pod is initialized in a 0.44m x 0.35m box (as in MimicGen [11]).

There are two skill segments: pick pod and both insert pod into machine as well as close lid. Here, the insertion and closing are treated as a single learned skill.

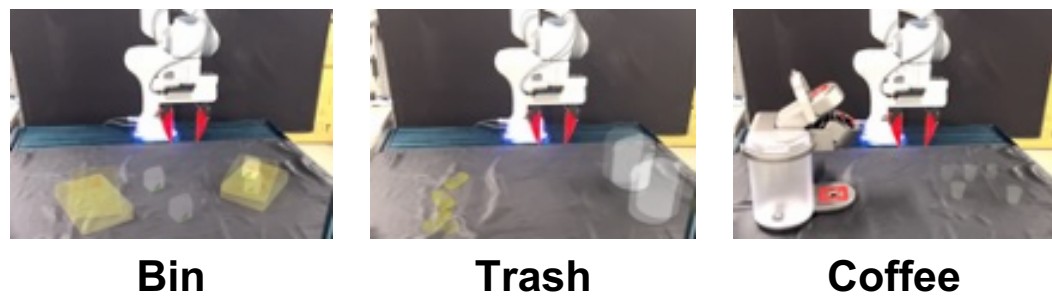

**Bin**          **Trash**          **Coffee**

Figure J.3: **Real-World Reset Distributions.** The initial states of the Pick-Place-Milk, Cleanup-Butter-Trash, and Coffee tasks each overlaid onto a single image.

# K  Sim-to-Real Experiment

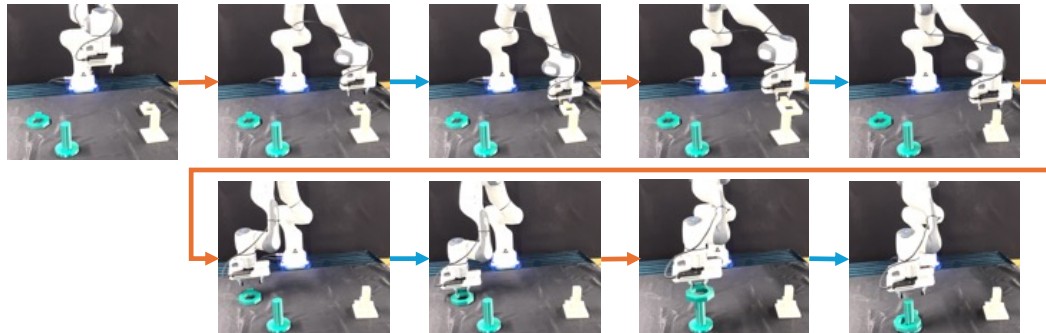

Figure K.1: **Real-World Nut Assembly Execution.** An example execution of the real-world Nut Assembly task. The task involves four skill segments (in blue) and four motion planning segments (in orange). The skill segments are 1) pick the Square Nut, 2) place the Square Nut on the Square Peg, 3) pick the Round Nut, 4) place the Round Nut on the Round Peg.

As described in Section 6.3, we performed an experiment to explore SkillGen's ability at facilitating zero-shot sim-to-real transfer. In this section, we provide further details omitted in the main text. We considered the "Nut Assembly" task (Figure K.1) where the robot must pick a Square Nut, place the Square Nut on the Square Peg, pick a Round Nut, and place the Round Nut on the Round Peg. This task is long-horizon in that it involves four skill stages; additionally, each place stage require precise manipulation to fit each nut on its associated peg.

**Insertion Tolerance.** We designed a simulated analog of the task, "Nut Assembly [Sim]", in robosuite [78]. In the real world, we replicate the CAD model of each nut and peg and 3D printed them so that the real and simulated geometries matched. The square peg is 3.2cm on each side, the square hole on the nut is 4.6cm on each side, the round peg is 4cm in diameter, and the round hole is approximately 6.8cm in diameter, leaving only a couple centimeters of tolerance for each nut insertion.

**Initialization Bounds.** In simulation, the nuts and pegs are each initialized randomly in non-overlapping 21cm x 41.5m quadrants of the table, with fixed orientation. In the real world, the initialization region for the objects are as follows: square nut (18cm x 26cm), round nut (20cm x 20cm), square peg (14cm x 40cm) and round peg (16cm x 30cm). The simulation bounds were intentionally designed to be more extensive than the real world initialization bounds.

**Observation and Action Space.** As mentioned in Section 7, we made several assumptions specifically for this experiment. First, we trained pose-based rather than image-based policies. As a result, there is no visual sim-to-real transfer. Second, because pose estimation during robot execution can be challenging, for example, due to the robot occluding the camera, each policy observes only the initial object poses. They do however consume up-to-date robot proprioception measurements, consisting of the end effector position and width of the gripper fingers. We make an additional simplification, and provide the end effector position with respect to the initial object position for all 4 items, instead of providing the end effector position and the object poses separately. Consequently, the final observation consumed by the agent is simply the robot end effector position with respect to the initial square nut position, square peg position, round nut position, and round peg position, as well as the width of the gripper fingers. Additionally, we simplified the agents action space by fixing the orientation of the end effector, which results in a 4-DOF position-only action space (one extra dim for gripper actuation).

**Policy Training Details.** We mostly follow the procedure described in Appendix H for HSP-Class training and the procedure from MimicGen [11] for training the MimicGen policies. We use an increased learning rate of 1e-3 for the closed-loop policy network. We also change the RNN policy to make it "open-loop" over the RNN horizon by repeating the first observation in the sequence instead of providing the current observation – this is equivalent to the action chunking described in Zhao et al. [19].

**Experiment Summary.** Ultimately, through SkillGen, we were able amplify a single simulation source demonstration into 1000 simulation demonstrations on Nut Assembly [Sim], train a pose-

based HSP-Class policy, and deploy it using SkillGen without any real-world data, where it achieved 35% success rate, while the MimicGen agent could not complete the full task, and achieved 5% success rate on the first square nut insertion.

# L  Results with Conventional Teleoperation Source Demonstrations

| Task Variant | MimicGen [11] | SkillGen |
|---|---|---|
| Square ($D_0$) | 73.7 | **87.3** |
| Square ($D_1$) | 48.9 | **73.8** |
| Square ($D_2$) | 31.8 | **65.1** |
| Threading ($D_0$) | **51.0** | 43.6 |
| Threading ($D_1$) | **39.2** | 36.7 |
| Threading ($D_2$) | 21.6 | **36.9** |
| Piece Assembly ($D_0$) | 35.6 | **48.7** |
| Piece Assembly ($D_1$) | 35.5 | **48.4** |
| Piece Assembly ($D_2$) | 31.3 | **53.8** |
| Coffee ($D_0$) | 78.2 | **81.5** |
| Coffee ($D_1$) | 63.5 | **75.4** |
| Coffee ($D_2$) | 27.7 | **59.8** |
| Average | 44.8 | **59.3** |

Table L.1: **Data Generation Rates from using Conventional Teleoperation Source Data.** SkillGen improves data generation rates over MimicGen substantially for most tasks, particularly the $D_2$ variants.

| Task Variant | MimicGen [11] | HSP-Class | HSP-Reg |
|---|---|---|---|
| Square ($D_0$) | 90.7 | **100.0** | 84.0 |
| Square ($D_1$) | 73.3 | **84.0** | 58.0 |
| Square ($D_2$) | 49.3 | **68.0** | 46.0 |
| Threading ($D_0$) | **98.0** | 94.0 | 94.0 |
| Threading ($D_1$) | **60.7** | 46.0 | 56.0 |
| Threading ($D_2$) | 38.0 | 34.0 | **50.0** |
| Piece Assembly ($D_0$) | **82.0** | 80.0 | 74.0 |
| Piece Assembly ($D_1$) | **62.7** | 48.0 | 52.0 |
| Piece Assembly ($D_2$) | 13.3 | **42.0** | 36.0 |
| Coffee ($D_0$) | **100.0** | 98.0 | **100.0** |
| Coffee ($D_1$) | 90.7 | **100.0** | 94.0 |
| Coffee ($D_2$) | 77.3 | **92.0** | 90.0 |
| Average | 70.0 | **73.8** | 69.5 |

Table L.2: **Agent Performance on Datasets Generated from Conventional Teleoperation Source Data.** Across the tasks, the average SkillGen policy learning results are comparable to MimicGen, but HSP-Class slightly outperforms the MimicGen baseline.

The experiments presented in Sec. 6 used demonstrations collected with HITL-TAMP [13], a teleoperation system where humans only demonstrate select skill segments of each task. A TAMP system plans and executes the rest of the task, in between skill demonstrations. In this section, we analyze how SkillGen's performance changes when using source demonstrations from a conventional teleoperation system instead of HITL-TAMP.

We use the same source demonstrations as MimicGen, and annotate the skill phases (Sec. 4) to enable data generation with SkillGen. Table L.1 shows that the average data generation rate is higher by 15% over MimicGen. However, Table L.2 shows that the average policy learning results are comparable to MimicGen (compared to the substantial improvements over MimicGen from using HITL-TAMP source data in Fig. 4). This shows that SkillGen performance is higher when using HITL-TAMP source data. One potential reason is due to the variability in motion planner poses when using manual annotations compared to the consistent annotations based on pre-conditions coming from the HITL-TAMP system. This variability can pose a challenge for learning methods [61]. Analyzing this gap further is a valuable avenue for future work.

# M    Ablations

| Task Variant | H-TAMP | H-TAMP(+T) | H-Class | H-Class(-T) | H-Reg | H-Reg(-T) | H-Reg(-R) |
|---|---|---|---|---|---|---|---|
| Square ($D_0$) | **100.0** | **100.0** | **100.0** | **100.0** | 94.0 | 98.0 | 80.0 |
| Square ($D_2$) | 94.0 | 90.0 | 94.0 | **96.0** | 52.0 | 46.0 | 40.0 |
| Threading ($D_0$) | **100.0** | **100.0** | 92.0 | 94.0 | 94.0 | 92.0 | **100.0** |
| Threading ($D_1$) | **72.0** | 68.0 | 66.0 | 58.0 | 60.0 | 66.0 | 58.0 |
| Piece Assembly ($D_0$) | **96.0** | 94.0 | 80.0 | 80.0 | 86.0 | 80.0 | 80.0 |
| Piece Assembly ($D_2$) | **84.0** | 82.0 | 74.0 | 76.0 | 50.0 | 40.0 | 14.0 |
| Coffee ($D_0$) | **100.0** | **100.0** | **100.0** | **100.0** | **100.0** | **100.0** | **100.0** |
| Coffee ($D_2$) | 94.0 | **100.0** | **100.0** | 98.0 | 98.0 | 96.0 | 56.0 |

Table M.1: **Ablation of Key Components.** To understand the difficulty of predicting policy termination, we modify HSP-TAMP to use a termination classifier (HSP-TAMP (+T)), and modify HSP-Class and HSP-Reg to use TAMP to handle termination instead of the termination classifier (-T variants). We see that performance is largely unchanged, indicating that learning termination is relatively easy. To understand the value of initiation augmentation (Sec. 4.5), we train HSP-Reg on dataset generated without it. The large performance regressions demonstrate it can be critical.

## M.1    Difficulty of Predicting Policy Termination

To understand the difficulty of predicting policy termination, we make the following changes to each method. HSP-TAMP (+T): we modify HSP-TAMP to use the same policy termination classifier $\mathcal{T}_\psi(o_t)$ from HSP-Class and HSP-Reg, and use it instead of TAMP to determine when agent $\pi_\theta$ should terminate and cede control back to TAMP. HSP-Class (-term) and HSP-Reg (-T): we use the same conditions as HSP-TAMP to dictate when to cede control from the agent $\pi_\theta$ back to the motion planner, instead of using the classifier. We see that the performance of HSP-TAMP (+T) is at most 4% below and 6% above HSP-TAMP, showing that predicting policy termination is not very difficult. Comparing HSP-Class (-term) to HSP-Class (8% lower to 2% higher) and HSP-Reg (-T) to HSP-Reg (10% lower to 6% higher) corroborates this claim. By comparison, the significant difference in performance between HSP-Class and HSP-Reg on a select few tasks (analyzed in Sec. 6) demonstrates that motion planner target prediction is significantly more challenging. This suggests that the key bottleneck for improving HSP-Reg performance is improving its ability to predict motion planner target poses – there is consequently an opportunity for future work to improve this by integrating models that utilize 3D information [58, 59] or exploring alternative model architectures. See Appendix Q for further discussion.

## M.2    Value of Initiation Augmentation

To show the value of initiation augmentation (Sec. 4.5), we train HSP-Reg on datasets generated without motion augmentation (HSP-Reg (-R)) and compare with HSP-Reg. Removing motion augmentation can cause significant performance drops (e.g. 40% drop on Coffee $D_2$, 26% on Three Piece Assembly $D_2$), showing that it can be critical to enable better performance by allowing agents to recover from incorrect motion planner target predictions.

# N Robot Transfer

We apply SkillGen to generate datasets and train agents for a robot arm that is different than the one the human collected source demonstrations on (Fig. N.1). We use the same source demonstrations as those used in our main experiments, collected on the Panda arm, and generate demonstrations for the Sawyer arm. The results are presented in Table N.1 (data generation) and Table N.2 (policy learning). We see that data generation rates are substantially higher for SkillGen than MimicGen, and that HSP-Class policies trained on SkillGen data are higher performing than their MimicGen counterparts.

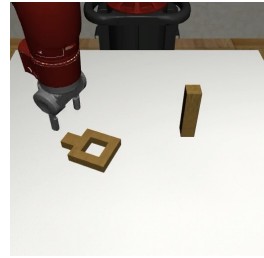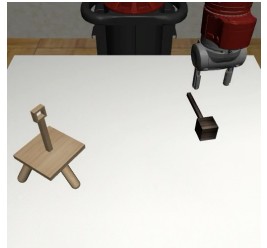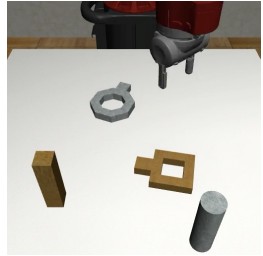

Figure N.1: **Data Generation for Sawyer Robot Arm.** Example configurations for task variants where Skill-Gen generated data for the Sawyer robot arm, using source human data collected on the Panda robot arm.

| Task Variant | MimicGen [11] | SkillGen |
|---|---|---|
| Square ($D_0$) (Panda) | 73.7 | **99.8** |
| Square ($D_0$) (Sawyer) | 55.8 | **95.2** |
| Square ($D_1$) (Panda) | 48.9 | **91.5** |
| Square ($D_1$) (Sawyer) | 38.8 | **94.0** |
| Threading ($D_0$) (Panda) | 51.0 | **76.2** |
| Threading ($D_0$) (Sawyer) | 28.8 | **68.2** |
| Threading ($D_1$) (Panda) | 39.2 | **66.4** |
| Threading ($D_1$) (Sawyer) | 23.7 | **62.5** |
| Nut Assembly ($D_0$) (Panda) | 53.0 | **98.6** |
| Nut Assembly ($D_0$) (Sawyer) | 34.7 | **86.1** |
| Nut Assembly ($D_1$) (Panda) | 30.0 | **91.7** |
| Nut Assembly ($D_1$) (Sawyer) | 22.1 | **78.3** |

Table N.1: **Data Generation Rates for Generating Datasets for Different Robots.** We use SkillGen to produce datasets on the Sawyer robot arm using the same 10 source demos collected on the Panda arm. SkillGen improves data generation rates substantially over MimicGen.

| Task Variant | MimicGen [11] | HSP-Class |
|---|---|---|
| Square ($D_0$) (Panda) | 90.7 | **100.0** |
| Square ($D_0$) (Sawyer) | 86.0 | **96.0** |
| Square ($D_1$) (Panda) | 73.3 | **98.0** |
| Square ($D_1$) (Sawyer) | 60.7 | **98.0** |
| Threading ($D_0$) (Panda) | **98.0** | 92.0 |
| Threading ($D_0$) (Sawyer) | 88.7 | **94.0** |
| Threading ($D_1$) (Panda) | 60.7 | **66.0** |
| Threading ($D_1$) (Sawyer) | 50.7 | **54.0** |
| Nut Assembly ($D_0$) (Panda) | 60.0 | **92.0** |
| Nut Assembly ($D_0$) (Sawyer) | 74.0 | **88.0** |
| Nut Assembly ($D_1$) (Panda) | 16.0 | **78.0** |
| Nut Assembly ($D_1$) (Sawyer) | 8.0 | **62.0** |

Table N.2: **Agent Performance on Generated Datasets for Different Robot Arms.** We use SkillGen to produce datasets on the Sawyer robot arm using the same 10 source demos collected on the Panda arm. HSP-Class policies trained on SkillGen data significantly outperform agents trained on MimicGen data.

# O  Algorithm Pseudocode

Algorithm 1 provides the pseudocode for the data generation process described in Section 4.4. For each skill trajectory in the source demonstration, SkillGen first estimates the current pose of the object that the skill manipulates. This is used to transform the stored initiation state. Then, MOTION-PLANNER solves for a robot configuration that reaches this pose and plans a joint-space path to the configuration, executed with a joint space controller. Finally, each end-effector action is adapted to the world frame and executed using task-space control.

---
**Algorithm 1** Demonstration Generation
---

**procedure** GENERATE-DATA($\tau$)
    **for** $\tau_{is} \in \tau$ **do**
        $T_W^{O'_i} \leftarrow$ ESTIMATE-POSE()
        $T_W^{E'_0} \leftarrow T_W^{O_i} \tau_{is}[0]$
        $q_0 \leftarrow$ CURRENT-CONFIG()
        $q_* \leftarrow$ INVERSE-KINEMATICS($T_W^{E'_0}$)
        **for** $q \in$ MOTION-PLANNER($q_0, q_*$) **do**
            JOINT-SPACE-CONTROL($q$)
        **for** $T_{O_i}^{E_t} \in \tau_{is}$ **do**
            $T_W^{E'_t} \leftarrow T_W^{O'_i} T_{O_i}^{E_t}$
            TASK-SPACE-CONTROL($T_W^{E'_t}$)

---

Algorithm 2 provides the pseudocode for HSP deployment, which was described Section 4.6. The structure has some global similarity with Algorithm 1, but critically, it operates over skills instead of trajectories and does not require pose estimation. For each skill in a provided sequence of skills $\Psi$, DEPLOY-HSP predicts the initiation pose using the current observation $o$. Then, it plans and executes joint-space motions to the initiation pose. Until the termination network predicts to terminate, the skill queries its policy for the next task-space action.

---
**Algorithm 2** HSP Deployment
---

**procedure** DEPLOY-HSP($\Psi$)
    **for** $\langle O, \mathcal{I}_\theta, \pi_\theta, \mathcal{T}_\theta \rangle \in \Psi$ **do**
        $o \leftarrow$ OBSERVE()
        $T_W^{E'_0} \leftarrow \mathcal{I}_\theta(o)$
        $q_0 \leftarrow$ CURRENT-CONFIG()
        $q_* \leftarrow$ INVERSE-KINEMATICS($T_W^{E'_0}$)
        **for** $q \in$ MOTION-PLANNER($q_0, q_*$) **do**
            JOINT-SPACE-CONTROL($q$)
        **while** $\mathcal{T}_\theta(o) \neq$ **True do**
            $T_W^{E'_t} \leftarrow \pi_\theta(o)$
            TASK-SPACE-CONTROL($T_W^{E'_t}$)
            $o \leftarrow$ OBSERVE()

---

# P    Comparison with HITL-TAMP

As we described in Section 2, HITL-TAMP [13] is a prior system that integrates BC and planning to improve both data collection efficiency and policy success rates. Within SkillGen, we optionally use HITL-TAMP to both collect a handful of source demonstrations (Section 4.3) and deploy learned skills at test time through HSP-TAMP (Section 4.6). However, when compared directly, SkillGen has several advantages over HITL-TAMP.

**Fewer Assumptions.** HITL-TAMP requires a model to plan the TAMP segments. Specifying one requires defining Planning Domain Definition Language (PDDL) actions, including their parameters, preconditions, and effects, along with sampling procedures that generate continuous action parameter values [91]. In contrast, SkillGen only requires a skill plan at data generation time, namely the sequence of objects that will be acted upon. At test time, SkillGen can even reduce this assumption by learning a single skill that encompasses all learned segments without explicitly conditioning on any objects. HITL-TAMP also requires pose observation or estimation during all its phases, for example, to define TAMP-gated hand-off regions from TAMP to a learned policy. Through directly learning initiation sets, which can be viewed as learning-gated conditions, SkillGen not only uses learning to transfer control but also avoids pose estimation in its HSP-Reg configuration.

**Lower Human Effort.** Although HITL-TAMP partially automates the demonstration process, a human must still manually teleoperate a portion of each episode. Thus, the amount of human effort required scales linearly with the number of demonstrations. In contrast, SkillGen only requires a fixed amount of human effort and can spawn an arbitrarily large number of demonstrations. Furthermore, policy learning results can be comparable given a similar amount of SkillGen demonstrations and HITL-TAMP demonstrations (Sec. 6.2), with just a fraction of the human effort.

**Object Grasp Segments Delegated to Learned Policies.** HITL-TAMP, TAMP is responsible for carrying out object grasps, while SkillGen defers all object interaction to learned agents. For example, in the real-world Coffee task, TAMP controls the arm to grasp the coffee pod and approach the insertion point on the machine, and a learned policy is only responsible for the insertion segment. In SkillGen, there are two skill segments that must be learned – one for pod grasping and one for pod insertion. Despite this increased difficulty, a policy trained on SkillGen data performs comparably. HSP-Class obtains 65% on the Coffee task with 100 SkillGen demos generated from just 3 human demos, in comparison to HSP-TAMP achieving 74% from 100 HITL-TAMP demos.

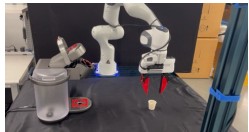 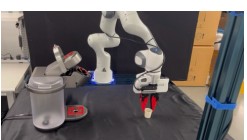 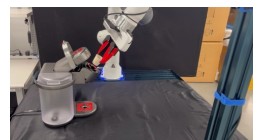 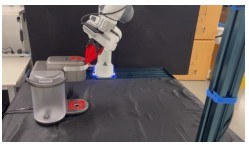

Figure P.1: **Comparison Between Skill Segments Learned by SkillGen and HITL-TAMP.** The experiments in HITL-TAMP [13] assumed that TAMP carries out object grasps (the left two frames for the Coffee task shown above) – consequently, the trained agent was responsible for less portions of each task (e.g. the right two frames for the Coffee task above). By contrast, SkillGen is responsible for all segments shown above.

# Q  Discussion on HSP-Reg Results

HSP-Reg makes the fewest assumptions out of the three HSP methods presented in this work (Sec. 4.6). However, while the average task success rate is only lower by 10% to 13% than the other methods, there can still be a significant gap in policy performance depending on the specific task. In this section, we provide some reasons to be optimistic that HSP-Reg performance can be increased significantly.

**Using more SkillGen data.** In this work, our main experiments (Fig. 4) used 1000 SkillGen demonstrations – this number was chosen for consistency with prior work [11]. However, we found that using more demonstrations can significantly boost HSP-Reg results (Appendix E). Some notable performance increases from 1000 SkillGen demos to 5000 SkillGen demos include Square $D_2$ (52% to 72% on HSP-Reg) and Threading $D_1$ (60% to 76% on HSP-Reg).

**Improving agent observability.** HSP-Reg is responsible for directly predicting a 6-DoF target pose for the motion planner to reach – this can be the key bottleneck for improving performance (corroborated by ablations in Appendix M and the performance gap between HSP-Reg and HSP-Class). This can be a difficult prediction problem when using just a front-view and wrist-view image for this prediction. Consequently, we ran an experiment to see if adding a third, side-view image would improve results. We used 5000 SkillGen demos with front-view, wrist-view, and side-view observations, and obtained our best HSP-Reg results – 82% for Square $D_2$ (compared to the 52% in Fig. 4), 86% for Threading $D_1$ (compared to 60%), and 74% for Piece Assembly $D_2$ (compared to 50%). These results also demonstrate that adding depth information for the pose prediction can be beneficial, as used in prior work [58, 59].

# R  Skill Segments and Annotations

In order to amplify a set of source demonstrations in a targeted manner, SkillGen requires annotation of the start and end of each skill that should be learned on the demonstrations. Even when using HITL-TAMP to gather source demonstrations, the TAMP model must specify action preconditions and effects (Appendix P), which loosely correspond to skill initiation and termination conditions.

The choice of what skills to learn and how fine-grained they should be can be customized by a human supervisor. For example, in the Coffee task displayed Fig. J.2, the robot must insert the pod and then close the coffee machine lid. We choose to model and learn both behaviors as a single skill rather than split them into two separate skills, connected by transit motion planning. This imposes a larger burden on learning but reduces the execution time by not requiring motion planning between the two behaviors.

Ultimately, the motivation of Sec. 4 is our primary recommendation with respect to modeling principles. Motion planning is a safe and reliable technique for addressing contact-adverse segments of tasks, which are often substantial in many common tasks. If learned policies are able to replicate these attributes for a given task, then it makes sense to incorporate more learning. Otherwise, deferring learning to primarily the contact-rich task segments, where motion planning is ineffective is the wiser strategy.

# S Comparison with Replay-Noise Baseline

| Task | Replay-Noise | SkillGen |
|------|-------------|----------|
| Square ($D_0$) | **99.8** | **99.8** |
| Threading ($D_0$) | **76.1** | **76.2** |
| Piece Assembly ($D_0$) | **82.3** | **82.5** |
| Coffee ($D_0$) | **74.2** | **73.3** |

Table S.1: **Data Generation Results on Replay-Noise baseline.** SkillGen achieves comparable data generation success rates compared to a baseline that replays the source demonstrations and adds action noise.

| Task Variant | HSP-T (RN) | HSP-T (SG) | HSP-C (RN) | HSP-C (SG) |
|---|---|---|---|---|
| Square $D_0$ | 82.0 | **100.0** | 80.0 | **100.0** |
| Square $D_1$ | 24.0 | **100.0** | 6.0 | **98.0** |
| Square $D_2$ | 12.0 | **94.0** | 4.0 | **94.0** |
| Threading $D_0$ | **100.0** | **100.0** | 88.0 | 92.0 |
| Threading $D_1$ | 2.0 | **72.0** | 4.0 | 66.0 |
| Threading $D_2$ | 0.0 | **62.0** | 0.0 | 50.0 |
| Piece Assembly $D_0$ | 64.0 | **96.0** | 70.0 | 80.0 |
| Piece Assembly $D_1$ | 78.0 | **88.0** | 10.0 | 78.0 |
| Piece Assembly $D_2$ | 46.0 | **84.0** | 2.0 | 74.0 |
| Coffee $D_0$ | **100.0** | **100.0** | 100.0 | 100.0 |
| Coffee $D_1$ | 8.0 | **100.0** | 26.0 | **100.0** |
| Coffee $D_2$ | 0.0 | 94.0 | 0.0 | **100.0** |
| **Average** | 43.0 | **90.8** | 32.5 | 86.0 |

Table S.2: **Agent Performance on Datasets Generated by Replay-Noise Baseline.** This table compares the performance of agents trained on datasets generated by the Replay-Noise baseline, which replays the source demonstrations with action noise, to agents trained on SkillGen datasets. Policies trained on SkillGen $D_0$ datasets are more proficient than those trained on the Replay-Noise datasets. Furthermore, Replay-Noise is unable to generate data for $D_1$ and $D_2$ as the source demonstrations do not cover this distribution. Consequently, agents trained on Replay-Noise data perform poorly on the other task variants that are unseen during data generation.

We compare SkillGen against another data generation baseline that replays the source demonstrations with action noise – we call this baseline **Replay-Noise**. Concretely, we start with the same set of source demonstrations. At the start of each data generation attempt, we select a random source demonstration, and reset the simulator state to the initial state in the demonstration. We then replay the actions from the demonstration with the same level of action noise (0.05) used in our experiments, and keep the executed trajectory if it is successful. Note that motion segments in the source demonstrations are ignored (as in SkillGen) and a motion planner is used to directly reach the starting robot configuration of each skill segment in the source demonstration. We continue data collection in this manner until 1000 successful trajectories are collected, for fair comparison with the results presented in the main text.

We present the data generation rates for the Replay-Noise baseline in Table S.1 and the success rates of HSP-TAMP and HSP-Class agents trained on the generated datasets in Table S.2, and compare against SkillGen. Note that this baseline can only generate data for the same reset distribution as the source demonstrations ($D_0$) since the task resets used for data collection come directly from the source data. Consequently, all agents trained on Replay-Noise data are trained on the same dataset ($D_0$) but evaluated across all task variants. The results show that data generation rates on the source distribution ($D_0$) are similar, as expected, but policies trained on SkillGen data are more proficient on $D_0$ than those trained on replay data. Furthermore, SkillGen can train capable agents on reset distributions not explicitly collected by the human ($D_1$, $D_2$) unlike the Replay-Noise baseline, where agents achieve lower, often near-zero success rates due to the inability to collect relevant data.

This comparison shows the value of using SkillGen to adapt existing human demonstrations and collect demonstrations on new task instances (Sec. 4.4), instead of just replaying the existing demonstrations with noise.

# T    Results Across Multiple Seeds

| Task Variant | Src | MG | HSP-T | HSP-C | HSP-R |
|---|---|---|---|---|---|
| Square ($D_0$) | $60.7 \pm 7.7$ | $90.7 \pm 1.9$ | $\mathbf{100.0 \pm 0.0}$ | $\mathbf{100.0 \pm 0.0}$ | $94.0 \pm 3.3$ |
| Square ($D_1$) | - | $73.3 \pm 3.4$ | $\mathbf{99.3 \pm 0.9}$ | $97.3 \pm 0.9$ | $70.7 \pm 6.6$ |
| Square ($D_2$) | - | $49.3 \pm 2.5$ | $\mathbf{94.7 \pm 4.1}$ | $91.3 \pm 1.9$ | $53.3 \pm 1.9$ |
| Threading ($D_0$) | $56.7 \pm 9.0$ | $\mathbf{98.0 \pm 1.6}$ | $\mathbf{98.0 \pm 1.6}$ | $96.0 \pm 3.3$ | $95.3 \pm 1.9$ |
| Threading ($D_1$) | - | $60.7 \pm 2.5$ | $\mathbf{70.0 \pm 2.8}$ | $62.0 \pm 5.7$ | $63.3 \pm 2.5$ |
| Threading ($D_2$) | - | $38.0 \pm 3.3$ | $63.3 \pm 0.9$ | $54.7 \pm 5.2$ | $\mathbf{64.7 \pm 3.8}$ |
| Piece Assembly ($D_0$) | $50.7 \pm 16.4$ | $82.0 \pm 1.6$ | $\mathbf{94.0 \pm 2.8}$ | $80.0 \pm 3.3$ | $82.7 \pm 3.4$ |
| Piece Assembly ($D_1$) | - | $62.7 \pm 2.5$ | $\mathbf{89.3 \pm 0.9}$ | $78.0 \pm 1.6$ | $68.7 \pm 0.9$ |
| Piece Assembly ($D_2$) | - | $13.3 \pm 3.8$ | $\mathbf{84.7 \pm 0.9}$ | $74.7 \pm 0.9$ | $48.7 \pm 8.2$ |
| Coffee ($D_0$) | $\mathbf{99.3 \pm 0.9}$ | $\mathbf{100.0 \pm 0.0}$ | $\mathbf{100.0 \pm 0.0}$ | $\mathbf{100.0 \pm 0.0}$ | $\mathbf{100.0 \pm 0.0}$ |
| Coffee ($D_1$) | - | $90.7 \pm 2.5$ | $\mathbf{100.0 \pm 0.0}$ | $98.7 \pm 0.9$ | $98.0 \pm 1.6$ |
| Coffee ($D_2$) | - | $77.3 \pm 0.9$ | $\mathbf{97.3 \pm 2.5}$ | $97.3 \pm 1.9$ | $96.7 \pm 1.9$ |
| Nut Assembly ($D_0$) | $20.7 \pm 5.0$ | $63.3 \pm 3.4$ | $\mathbf{99.3 \pm 0.9}$ | $96.0 \pm 2.8$ | $88.0 \pm 8.5$ |
| Nut Assembly ($D_1$) | - | $18.0 \pm 4.3$ | $\mathbf{77.3 \pm 7.5}$ | $74.0 \pm 7.1$ | $6.7 \pm 9.4$ |
| Nut Assembly ($D_2$) | - | $16.0 \pm 4.3$ | $\mathbf{56.0 \pm 4.3}$ | $50.0 \pm 3.3$ | $19.3 \pm 3.4$ |
| Coffee Prep ($D_0$) | $5.3 \pm 3.4$ | $\mathbf{97.3 \pm 0.9}$ | $92.0 \pm 1.6$ | $88.7 \pm 2.5$ | $84.0 \pm 3.3$ |
| Coffee Prep ($D_1$) | - | $42.0 \pm 0.0$ | $50.0 \pm 4.3$ | $\mathbf{65.3 \pm 6.2}$ | $59.3 \pm 4.1$ |
| Coffee Prep ($D_2$) | - | - | $82.7 \pm 2.5$ | $74.7 \pm 2.5$ | $\mathbf{84.0 \pm 1.6}$ |
| **Average** | - | 59.6 | **86.0** | 82.1 | 71.0 |

Table T.1: **Agent Performance on Source and Generated Datasets across Multiple Seeds.**   Success rates of agents trained on source demonstrations (with HSP-TAMP), MimicGen [11] data (with BC-RNN [1]), and SkillGen data (with all HSP variants) across 3 seeds per run. SkillGen data greatly improves agent performance on $D_0$ compared to the source data, and SkillGen agents substantially outperform MimicGen agents, especially on more challenging task variants.

In the main text, we presented policy learning results using a single seed. In this section, we present policy learning results across 3 seeds for the core set of SkillGen datasets presented in Fig. 4 (left) and show that the results are extremely similar. Comparing the averages from the single-seed results against the results averaged across 3 seeds in Table T.1, we see very similar performances (59.1 vs. 59.6 for MimicGen, 85.7 vs. 86.0 for HSP-TAMP, 82.9 vs. 82.1 for HSP-Class, and 72.6 vs. 71.0 for HSP-Reg).

Note that the MimicGen paper [11] (Appendix T) showed that data generation has very little variance empirically across multiple seeds. We also found this to be true for SkillGen, and as a result, we do not present results across multiple seeds for data generation.

