# OpenReview forum: "SkillMimicGen: Automated Demonstration Generation for Efficient Skill Learning and Deployment"
_robot-learning.org/CoRL/2024/Conference — CoRL 2024_

### Official Review · Reviewer_cZDT · 2024-07-19

**Originality:** 3
**Technical Quality:** 4
**Clarity Of Presentation:** 4
**Potential Impact:** 3
**Recommendation:** 4
**Confidence:** 4

**Review:**

The concepts discussed are simple and intuitive, it makes sense to use motion planning where appropriate, and it’s good to see the effect of scale on success, even though no additional demonstrations are used. The paper is well written and easy to follow.

The downside is the novelty seems rather weak, only a small delta from the MimicGen work, though the results and analysis seem sufficient to show a good improvement. I feel this is a good method for the community to build from.

“enabling skills to combined” -> ‘’enabling skills to be combined”

“sequence at at test” -> “sequence at test”

“HSC-Reg agents; HSC-TAMP and HSC-Class” -> Meant to be HSP? Also in Figure 4 capture and Figure 4 top right.

I think your point is proven with a few select references, having large spans of citations for undisputed statements (e.g. Robot teleoperation [14-23], BC’s use in robot manipulation [33–45] and data augmentation for increasing dataset size [46–57]) is unnecessary.

Please try to use peer-reviewed sources where appropriate for references:

E.g:

[10] M. Dalal et al  is from Conference on Robot Learning 2023

[24] C. Chi is from Robotics: Science and Systems (RSS) 2024

**Quality Of The Limitations Section:**

3

**Questions For Rebuttal:**

See above for minor edits.

**Robotics Focus:**

4

**Summary Of Paper:**

This paper introduces SkillGen, a method for generating data from a small number of human demonstrations. This is done by recording the trajectory (end-effector poses) at key stages in manipulation tasks i.e. segments that require interaction with objects, and joining these with motion planning in scene variations. This paper also presents Hybrid Skill Policy (HSP), skills that are trained with behavioral cloning on the trajectory segments that interaction occurs. Skills are defined to have an initiation condition (SE(3) end-effector pose) and a termination condition {0-1} for start and stop conditions respectively, to determine when to switch between skills and off-the-shelf motion planners. These methods are shown on several tasks using a Franka arm, in simulation and with real world examples, including data collected only on the real platform and with transfer from sim2real. Results show better performance than an existing method (MimicGen), particularly for scene variations. Several options for learning initiation conditions (classifier, regression) were formulated to reduce reliance on task sequence assumptions.

**Summary Of Recommendation:**

My recommendation is this work is accepted. Though simple, I think the ideas are useful for others. -> Updated score after rebuttal.

---

### Official Review · Reviewer_Gd22 · 2024-07-20
**The paper reads well and the contribution is clear to this reviewer.**

**Originality:** 3
**Technical Quality:** 3
**Clarity Of Presentation:** 5
**Potential Impact:** 3
**Recommendation:** 3
**Confidence:** 3

**Review:**

Key Contributions:
- SkillGen segments human demonstrations into manipulation skills and adapts these skills to new contexts. This segmentation and adaptation allow for the efficient generation of demonstration datasets from a limited number of initial human demonstrations​

- The proposed HSP framework learns skill initiation, control, and termination from the generated datasets. This framework enables the sequencing of skills using motion planning at test time, significantly improving the policy learning process​

- SkillGen demonstrates substantial improvements over existing data generation frameworks like MimicGen. The paper reports a 24% increase in success rates for agents trained on SkillGen data, highlighting the robustness of the system in varied and cluttered environments

- The system’s efficacy is validated through extensive experiments, including over 24,000 demonstrations generated from just 60 human demonstrations across 18 task variants in simulation. Additionally, SkillGen is successfully applied to real-world tasks and demonstrates zero-shot sim-to-real transfer capabilities​

**Quality Of The Limitations Section:**

2

**Questions For Rebuttal:**

The authors compared SkillGen only to MimicGen, was wondering if MimicGen is the only SOTA?

**Robotics Focus:**

4

**Summary Of Paper:**

The paper presents SkillGen, an innovative system designed to address the challenges of data generation in robot manipulation tasks through automated demonstration generation. The primary motivation is to mitigate the high costs and extensive human effort associated with collecting large datasets for training robots, particularly for complex, long-horizon tasks.

**Summary Of Recommendation:**

I think this is a decent paper for CoRL.

---

### Official Review · Reviewer_LdNH · 2024-07-22
**Good extension paper**

**Originality:** 3
**Technical Quality:** 4
**Clarity Of Presentation:** 5
**Potential Impact:** 3
**Recommendation:** 3
**Confidence:** 3

**Review:**

SkillGen effectively builds upon MimicGen by introducing a modular approach to demonstration generation. It then creates a whole system around this change, that enables varied demo generation, hybrid policies mixing learned and hand-crafted motions, etc. to show how well it really works. The decomposition of demonstrations into motion planning and learnable skills is a valuable contribution, enabling flexible and adaptable policy creation. The treatment of the problem using the options framework helps understand the clear transitions between motion planning to learned skills. The authors thoroughly evaluate their method across various tasks in sim and real, showcasing its potential for enhancing imitation learning.

While the paper presents a strong foundation, some areas could be strengthened. First and foremost, the current comparisons are only with MimicGen and all-human demos. A more comprehensive comparison to state-of-the-art methods in data-efficient imitation learning could be made. Example baselines are:
1. Adding noise to the input human demos to generate more demonstrations ( ‘noisy-demo’ baseline).
1. Getting demos from a VLM, given a task description: e.g. DiffGen[4], ManipulateAnything (VLM baseline)
1. Methods like NOD-TAMP[3] that can learn a skill from a single demonstration.

Adding such comparisons will provide a clearer picture of SkillGen's contributions. Additionally, exploring connections to other efficient learning from demonstration (LfD) approaches, like RLPD[2], and [1], could offer valuable insights. So would methods that involve self-play for robotics tasks.

The assumption of known object poses limits the generalizability of the method. Addressing this limitation, would significantly broaden the applicability of SkillGen, although it may be out of scope for the current submission. Furthermore, while the paper highlights the benefits of motion planning, it's crucial to clearly differentiate the novel contributions of SkillGen from the capabilities inherent to motion planning itself.

*References:*

[1] Rigter, M., Lacerda, B., & Hawes, N. (2020). A framework for learning from demonstration with minimal human effort. IEEE Robotics and Automation Letters, 5(2), 2023-2030.
[2] Ball, P. J., Smith, L., Kostrikov, I., & Levine, S. (2023, July). Efficient online reinforcement learning with offline data. In International Conference on Machine Learning (pp. 1577-1594). PMLR.
[3] Cheng, S., Garrett, C., Mandlekar, A., & Xu, D. (2023). NOD-TAMP: Multi-Step Manipulation Planning with Neural Object Descriptors. arXiv preprint arXiv:2311.01530.
[4] Jin, Y., Lv, J., Jiang, S., & Lu, C. (2024). DiffGen: Robot Demonstration Generation via Differentiable Physics Simulation, Differentiable Rendering, and Vision-Language Model. arXiv preprint arXiv:2405.07309.

**Quality Of The Limitations Section:**

3

**Questions For Rebuttal:**

These are stated in the feedback above -- primarily, it would help to place this work in the context of other works that are learning from little to no human input. Please get some comparisons and discussions with the other methods and baselines stated above.

**Robotics Focus:**

4

**Summary Of Paper:**

SkillGen, an extension of MimicGen, presents a system for generating diverse robotic demonstrations from limited human demos. It decouples motion planning and learnable skills, and creates a framework for hybrid policy generation. The paper demonstrates improved performance in both simulation and real-world scenarios compared to baselines.

**Summary Of Recommendation:**

Overall, SkillGen presents a promising approach to demonstration generation, but further comparison is needed to solidify its position within the broader landscape of efficient imitation learning.

---

### Author Rebuttal · Authors · 2024-08-09

Thanks to all the reviewers for their constructive feedback and time! We have attached a revised manuscript to this message where changes are shown in red, and also posted individual replies to each reviewer.

---

### Decision · Program_Chairs · 2024-09-04

**Decision:**

Accept

**Comment:**

All reviewers are very positive about the paper, and acknowledge its good empirical performance they uniformly recommend acceptance.